# Global mapping of oil palm planting year from 1990 to 2021

Adrià Descals[1], David L.A. Gaveau[2,3], Serge Wich[4], Zoltan Szantoi[5,6], and Erik Meijaard[7]

[1]CREAF, Cerdanyola del Vallès, Barcelona 08193, Spain
[2]TheTreeMap; Bagadou Bas, Martel 46600, France
[3]Visiting Professor, Jeffrey Sachs Center on Sustainable Development, Sunway University, 5, Jalan Universiti, Bandar Sunway, 47500 Petaling Jaya, Selangor
[4]School of Biological and Environmental Sciences, Liverpool John Moores University, Liverpool L3 3AF, United Kingdom
[5]Climate Action, Sustainability and Science Department, European Space Agency, Frascati 00044, Italy
[6]Stellenbosch University, Stellenbosch 7602, South Africa
[7]Borneo Futures, Bandar Seri Begawan, Brunei Darussalam

*Correspondence to*: Adrià Descals (a.descals@creaf.uab.cat)

**Abstract.** Oil palm is a controversial crop, primarily because it is associated with negative environmental impacts such as tropical deforestation. Mapping the crop and its characteristics, such as age, is crucial for informing public and policy discussions regarding these impacts. Oil palm has received substantial mapping efforts, but up-to-date accurate oil palm maps
for both extent and age are essential for monitoring impacts and informing concomitant debate. Here, we present a 10-meter resolution global map of industrial and smallholder oil palm, developed using Sentinel-1 data for the years 2016–2021 and a deep learning model based on convolutional neural networks. In addition, we used Landsat-5, -7, and -8 to estimate the planting year from 1990 to 2021 at a 30-meter spatial resolution. The planting year indicates the year of establishment for the current oil palm plantation, as of 2021, either newly planted or replanted oil palm in an existing oil palm plantation. We validated the
oil palm extent layer using 18,812 randomly distributed reference points. The accuracy of the planting year layer was assessed using field data collected from 5,831 industrial parcels and 1,012 smallholder plantations distributed throughout the global oil palm growing area. We found oil palm plantations covering a total mapped area of 23.98 Mha, and our area estimates are 16.82 ± 0.19 Mha of industrial and 7.37 ± 0.25 Mha of smallholder oil palm worldwide. The producer's and user's accuracy are 91.0 ± 2.5% and 91.8 ± 1.2% for industrial plantations, and 71.4 ± 0.7% and 72.4 ± 1.8% for smallholders, which improves upon a
previous global oil palm dataset, particularly in terms of omission of oil palm. The overall mean error between estimated planting year and field data was -0.24 years and the root-mean-square error was 2.65 years, but the agreement was lower for smallholders. Mapping the extent and planting year of smallholder plantations remains challenging, particularly for wild and sparsely planted oil palm, and future mapping efforts should focus on these specific types of plantations. The average oil palm plantation age was 14.1 years, and the area of oil palm over 20 years old was 6.28 Mha. Given that oil palm plantations are
typically replanted after 25 years, our findings indicate that this area will require replanting within the coming decade, starting from 2021. Our dataset provides valuable input for optimal land use planning to meet the growing global demand for vegetable oils. The global oil palm extent layer for the year 2021 and the planting year layer from 1990 to 2021 can be found at https://doi.org/10.5281/zenodo.13379129 (Descals, 2024).

# 1 Introduction

Vegetable oil crops cover around 543 million hectares (Mha), accounting for roughly 37% of the total land area dedicated to agricultural crop production across the globe (Meijaard et al., 2024). In other words, oil crops are a major agricultural land-use, and the expansion of land allocated to vegetable oil crops has outpaced that of other commodities. Among oil crops, oil palm produces the most oil in total volume, but on a relatively small area of land, 29.62 Mha in 2021 (FAO, 2022), or about one-quarter of the land area allocated to soybean, the second-most productive oil crop in total volume (Meijaard et al., 2024).

Oil palm, however, is a tropical crop, and its expansion over the past decades has resulted in the loss of tropical forest and associated high biological diversity (Meijaard et al., 2020). For example, oil palm replaced >4 Mha of primary forest from 2001 to 2022 in Indonesia and Malaysia, the world's largest producers (Gaveau et al., 2022). There has been much debate, often emotive and polarized, about the extent to which oil palm has contributed to deforestation and the loss of threatened wildlife (Candellone et al., 2023; Teng et al., 2020). Up-to-date information about oil palm planting locations is necessary to

inform this debate and clarify the extent of global deforestation caused by oil palm, which remains unknown.

In addition to identifying where oil palm was planted, knowing the year of planting is also important. The planting year allows for the estimation of the oil palm age, which is a factor that determines the palm's productivity (Corley and Tinker, 2008). Oil palm plantations are often cleared after 25 years and the land replanted with young palms because productivity declines after

that age (Ismail and Mamat, 2002). Furthermore, plantation age allows for the estimation of dendrometric variables such as biomass and height using allometric equations. Biomass and height are important for calculating the carbon stock and management costs of the plantation (Corley and Tinker, 2008; Tan et al., 2014). The clearing of older oil palms and their replacement with new ones entails both positive and negative impacts; it is expensive, has social implications (Fosch et al., 2023), can increase yield through better planting material, reduces ecological connectivity in a landscape (Ashton-Butt et al.,

2019), and also offers opportunities for restoration interventions (Wenzel et al., 2024). Thus, knowing the global extent and planting year of oil palm is valuable, and remote sensing serves as an important tool for obtaining this information.

Satellite remote sensing offers the capability to map both the extent and timing of oil palm development. Synthetic aperture radar (SAR) has been very useful for mapping the extent of oil palm because of the distinctive backscatter response of palm-like canopies. (Miettinen and Liew, 2011). This characteristic backscatter response in SAR data has enabled mapping the

global extent of closed-canopy oil palm stands using Sentinel-1 (Descals et al., 2021). To estimate the year of oil palm planting, previous studies have used satellite time series from MODIS (Xu et al., 2020) and Landsat (Danylo et al., 2021; Du et al., 2022; Gaveau et al., 2022). Their approach posits that the satellite time series can capture the various stages of oil palm development, in particular the moment when the land is cleared for oil palm planting. Similar studies have estimated the year

of tree cover loss (Hansen et al., 2013) and the timing of disturbances in primary forests (Vancutsem et al., 2021) by detecting land cover changes in the Landsat time series. Although the dataset presented in Descals et al., 2021 is the first comprehensive

global oil palm layer, its methodology consisted of a single-year classification that largely missed young oil palm and existing plantations that were replanted in previous years. A multi-annual oil palm classification using a longer Sentinel-1 time series can potentially reduce the omission of oil palm and provide a more accurate representation of the global oil palm extent.

Furthermore, the layer in Descals et al., 2021 only depicted the extent of oil palm as of 2019 and does not provide information on planting year. Using the Landsat time series, we can estimate the year of oil palm planting, a valuable source of information that can complement the oil palm extent layer.

This study presents a global oil palm extent layer at 10-meter resolution and a planting year layer at 30-meter resolution. For

the oil palm extent, we extended the 2019 oil palm classification presented in Descals et al., 2021 to the period 2016–2021 using Sentinel-1 data. This classification was based on a convolutional neural network that identified industrial and smallholder plantations. For the year of oil palm planting, we developed a methodology specifically designed to detect the early stages of oil palm development in the Landsat time series from 1990 to 2021.

## 2 Methods

### 2.1 Overview

The algorithm used in this study consisted of three parts (Fig. 1). The first part involved mapping the extent of oil palm plantations using a deep learning model that classified Sentinel-1 annual composites. We performed the classification on annual composites from 2016 to 2021, and merged the annual classifications to create a single layer that depicted the oil palm extent. The second step involved estimating the timing of land preparation for oil palm by performing a retrospective analysis

of the Landsat time series from 1990 to 2021. This step aimed at detecting the images in the Landsat time series that depicted the land preparation for oil palm. In the third step, we determined the planting year of the current oil palm plantation as the date that showed the lowest Normalized Difference Water Index (NDWI) value during the land preparation phase.

### 2.2 Mapping the extent of oil palm

### 2.2.1 Sentinel-1 compositing

Sentinel-1 is a synthetic-aperture radar satellite; it incorporates an active sensor in the C-band and provides scenes at a spatial resolution of 10 meters and at a revisit time of 6 days (Torres et al., 2012). The compositing approach employed in this study is the same as the one used for the 2019 oil palm layer (Descals et al., 2021). We used the single co-polarization, vertical transmit/vertical receive band (VV), and the dual-band cross-polarization, vertical transmit/horizontal receive band (VH) from the Ground Range Detected (GRD) product. We corrected the data for the local incident angle. The correction of Sentinel-1

data for the local incident angle uses SRTM Digital Elevation Data Version 4 to reduce terrain-induced variations in radar backscatter. The correction was applied to daily Sentinel-1 scenes. The code for the correction of the local incident angle and

the generation of the Sentinel-1 composites can be found in the Code availability section (Descals, 2021). The daily Sentinel-1 images were aggregated annually from 2016 to 2021 using the median for ascending and descending orbits separately. Temporal information, such as seasonal variations in spectral backscatter, was not extracted from the Sentinel-1 time series.

The final annual composites represent the average of these two orbit composites. Aggregating the orbits separately addresses imbalances in the number of scenes between orbits, which could otherwise introduce potential terrain-induced artifacts if one orbit prevails. Although Sentinel-1 data is available from 2014, systematic full coverage of the oil palm growing area started from 2016. For this reason, we generated the first annual oil palm map for 2016.

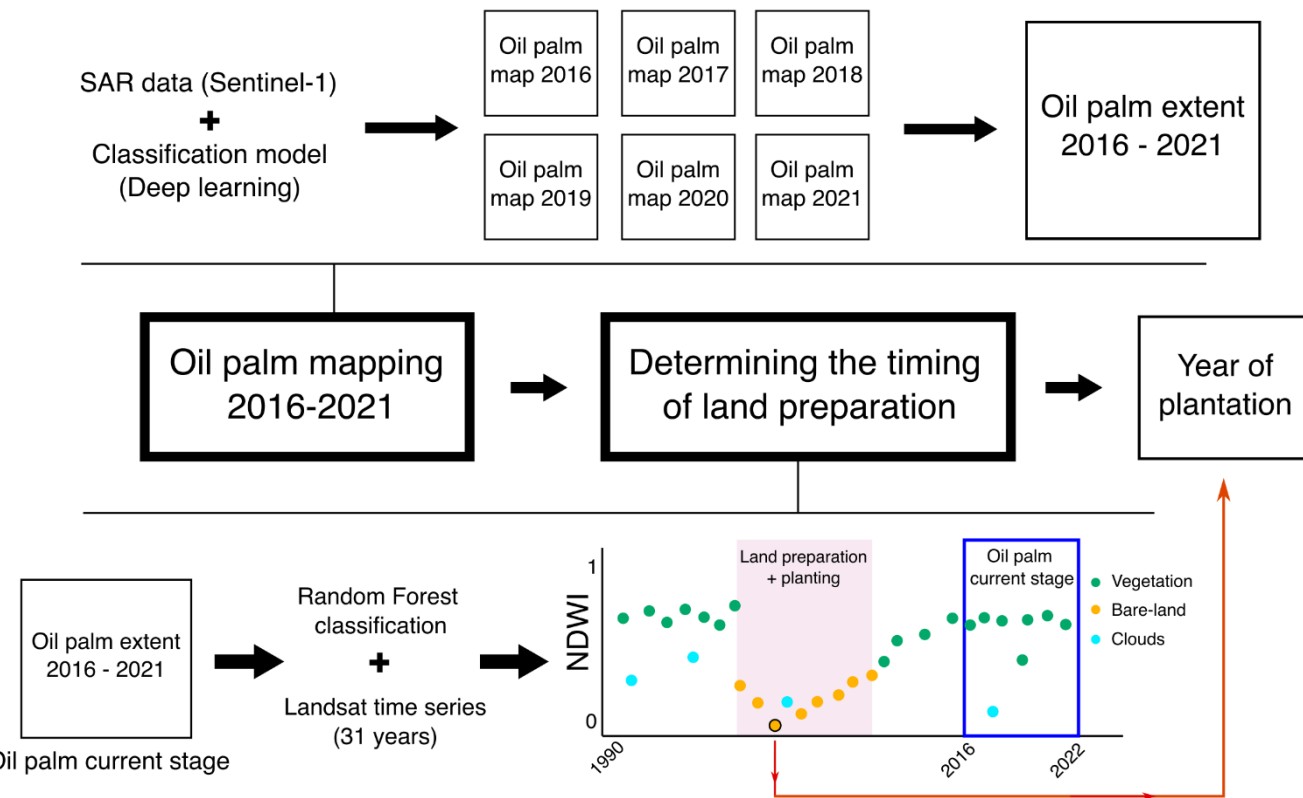


**Figure 1: Overview of the three main parts of the methodology for mapping the extent and planting year of oil palm.**

### 2.2.2 Deep learning classification

We used the deep learning model DeepLabv3+ to classify the Sentinel-1 composites into annual oil palm maps for the period 2016–2021. DeepLabv3+ is a supervised classification model for semantic segmentation based on convolutional neural

networks (Chen et al., 2017). The convolutional neural network architecture in this study mirrors that of the global oil palm layer for 2019 (Descals et al., 2021); the model underwent training using the same training dataset, comprising 296 training

images of 10 x 10 km. The model predicted three classes: Class 0: other land covers that are not closed-canopy oil palm; Class 1: closed-canopy industrial oil palm; and Class 2: closed-canopy smallholder oil palm.

In this study, we adopted the definitions of industrial and smallholder oil palm plantations from Descals et al., 2021. Industrial plantations typically span several thousand hectares, with uniform palm age and well-defined, often rectangular boundaries (Fig. A1). These plantations feature dense networks of roads or canals, designed during initial development of the plantation to optimize harvesting. On flat terrain, the roads are arranged in a rectilinear grid, while on hilly areas, they tend to curve. In contrast, smallholder plantations are usually less than 25 ha, though this threshold varies by country. Compared to industrial

plantations, smallholder plantations are less organized and have more diverse palm ages, forming a mosaic landscape mixed with other land uses. Large clusters of smallholder plantations have sparser trail networks than industrial ones.

One significant distinction between the 2019 oil palm layer, presented in Descals et al., 2021, and this work is the input data used in the deep learning model. For the global oil palm layer 2019, the classification model used as input the VV and VH

bands from Sentinel-1 GRD as well as the red band (B4) from Sentinel-2 Level-2A. Since Sentinel-2 Level-2A data were not available for the oil palm growing area before 2019, we excluded Sentinel-2 and used a classification model that only classified Sentinel-1 data. Since the original model required an input image with three channels, we stacked the VV and VH spectral images along with a third image filled with zeros. In this way, we could re-train the existing deep learning model without modifications in its architecture. One potential limitation of using only Sentinel-1 is that the classification model can increase

the commission error in other land cover types and, especially, in other palm species, such as coconut and sago palms. The false positives were already apparent in the 2019 global oil palm layer, with previous studies raising concerns about coconut plantations incorrectly classified as oil palm in Indonesia (Danylo et al., 2021; Descals et al., 2023). To address this issue, we applied two amendments to reduce the occurrence of false positives. First, we masked oil palm pixels that overlapped with the classes 'cropland', 'built-up', 'water bodies', 'herbaceous wetland', and 'mangrove' in the 10-m ESA WorldCover map v200

(Zanaga et al., 2022), given that oil palm is unlikely to be present in these land cover types. Second, we inspected the annual oil palm classification using high-resolution satellite imagery from Google Maps to remove any remaining false positives. We visually identified these false positives and reclassified them as the class 'other'.

### 2.2.3 Merging annual classifications 2016–2021

We combined the annual oil palm classification layers for the period 2016–2021 into a single layer that shows the extent of oil

palm. The merged oil palm layer depicts the oil palm detected at least once during the period 2016–2021. Since Sentinel-1 SAR scenes are inherently noisy, closed-canopy plantations are more likely to be missed in single-year classifications. For this reason, we did not provide the six annual classifications as a time series, but rather as a single layer depicting the situation as of 2021. We used three rules to merge the annual classification layers (Table 1). In the merged layer, a pixel equal to zero indicates the absence of oil palm throughout the 2016–2021 period. A pixel value of one indicates the detection of an industrial

plantation at least once in the 2016–2021 period, while a pixel value of two indicates the absence of industrial plantations and the detection of a smallholder plantation at least once in the same period.

Table 1: Rules for merging the annual oil palm classification maps (2016–2021) into the oil palm extent. The rules assign a class ('other', 'industrial oil palm', and 'smallholder oil palm') based on the same classes in the annual maps.

| Annual oil palm classification 2016-2021 | Oil palm extent 2016-2021 |
|---|---|
| All years are Class 'other' | Class 'other' |
| At least one year is Class 'industrial oil palm' | Class 'industrial oil palm' |
| At least one year is Class 'smallholder oil palm' and Class 'industrial oil palm' was absent | Class 'smallholder oil palm' |

The reclassification rules prioritize the presence of oil palm, especially industrial plantations, and classify a pixel as such if it detects oil palm in any year between 2016 and 2021. The following three arguments justify these rules:

1) Classification models using satellite data tend to underestimate the true oil palm area (Descals et al., 2021). For instance, the global oil palm layer 2019 showed an omission error that was substantially higher than the commission error. Considering these rates of omission and commission, oil palm is likely to be present in a pixel if the classification model has identified it at least once between 2016 and 2021.

2) These rules ensure that the merged layer reflects the replanting of oil palm plantations. For instance, a mature plantation that was clear-cut in 2017 would be detected as oil palm in 2016 but omitted in subsequent years (Fig. A2). By using these rules, such rotated plantations are included as oil palm despite being detected as 'other' in the years following the rotation.

3) The rules ensure that young plantations that reached full canopy closure during 2016–2021 are included as oil palm in the merged layer. This rule would affect young oil palm detected as 'other' in the first years of the period 2016–2021 but detected as oil palm once the plantation reaches full canopy closure within that timeframe.

### 2.2.4 Validation of the oil the palm extent

We conducted the validation of the global oil palm extent using the validation dataset developed for the 2019 global oil palm layer (Descals et al., 2021). This validation dataset included 10,816 points generated by simple random sampling and 2,679 points generated by stratified random sampling. In this study, we rejected the validation points generated by stratified random sampling because these points were sampled based on the mapped area in the 2019 oil palm layer and, thus, cannot be used

for validating the layer presented in this study. For the points generated by simple random sampling, we updated their true label considering the land cover for the period 2016–2021; in the original validation dataset, the true label only reflected the land cover scenario in 2019. Three cases arose in which the true label of a validation point required updating:

1) Young oil palm in 2019 that reached full canopy closure in 2020 and 2021. The true label was re-coded from class 'other' to class 'industrial oil palm' or 'smallholder oil palm'.

2) Closed-canopy oil palm that was clear-cut between 2016 and 2018. These plantations were interpreted as bare land (class 'other') in the 2019 validation dataset. The true label of these points was re-coded to 'industrial oil palm' or 'smallholder oil palm'.

3) Coconut plantations that were incorrectly interpreted as oil palm. The true label was re-coded to class 'other'. These points were mainly found in the Philippines and Indonesia. The recognition of coconut plantations was aided by an earlier study that mapped coconut palms at the global scale (Descals et al., 2023).

Additionally, we generated 7,148 new validation points across all grid cells where the merged oil palm layer included oil palm. After removing the stratified random sampling points from the original dataset, a high class imbalance became prominent, with few points labeled as industrial or smallholder oil palm. To address this, we added 1,000 new points generated through stratified random sampling: 700 for industrial oil palm and 300 for smallholder oil palm. The updated validation dataset includes a total of 18,812 points: 16,839 points were labeled with the class 'other', 1,374 points with the class 'industrial oil palm', and 531 points with the class 'smallholder oil palm'.

## 2.3 Determining the planting year with Landsat data

### 2.3.1 Landsat time series for determining the stages of oil palm development

The planting year was estimated using the Landsat time series (Landsat-5, -7, and -8). These optical satellites have provided 30-meter resolution images since 1984, depending on the region, and at a revisit time of 16 days each. We used the Landsat-5, -7, and -8 Level 2 Collection 2 Tier 1 products, which contain atmospherically corrected surface reflectance. Landsat can depict the different stages of oil palm, from the moment of land preparation to the young and maturity stages. To illustrate that, in Fig. 2, we present a time series of the Normalized Difference Water Index (NDWI) (Gao, 1996), calculated as the normalized difference between the near-infrared (band 5 in Landsat-8) and shortwave infrared bands (band 7 in Landsat-8), for an industrial oil palm plantation in South Papua, Indonesia. This time series reveals low NDWI values during land preparation for oil palm, followed by a steady increase during the growth of young oil palm. NDWI plateaus when the plantation reaches full canopy closure about three years after the planting. We selected NDWI because it is less noisy than indices relying on visible spectrum bands. NDWI uses SWIR and NIR bands, which can penetrate thin clouds and are less affected by atmospheric conditions like water vapor, which is typically high in the tropics.

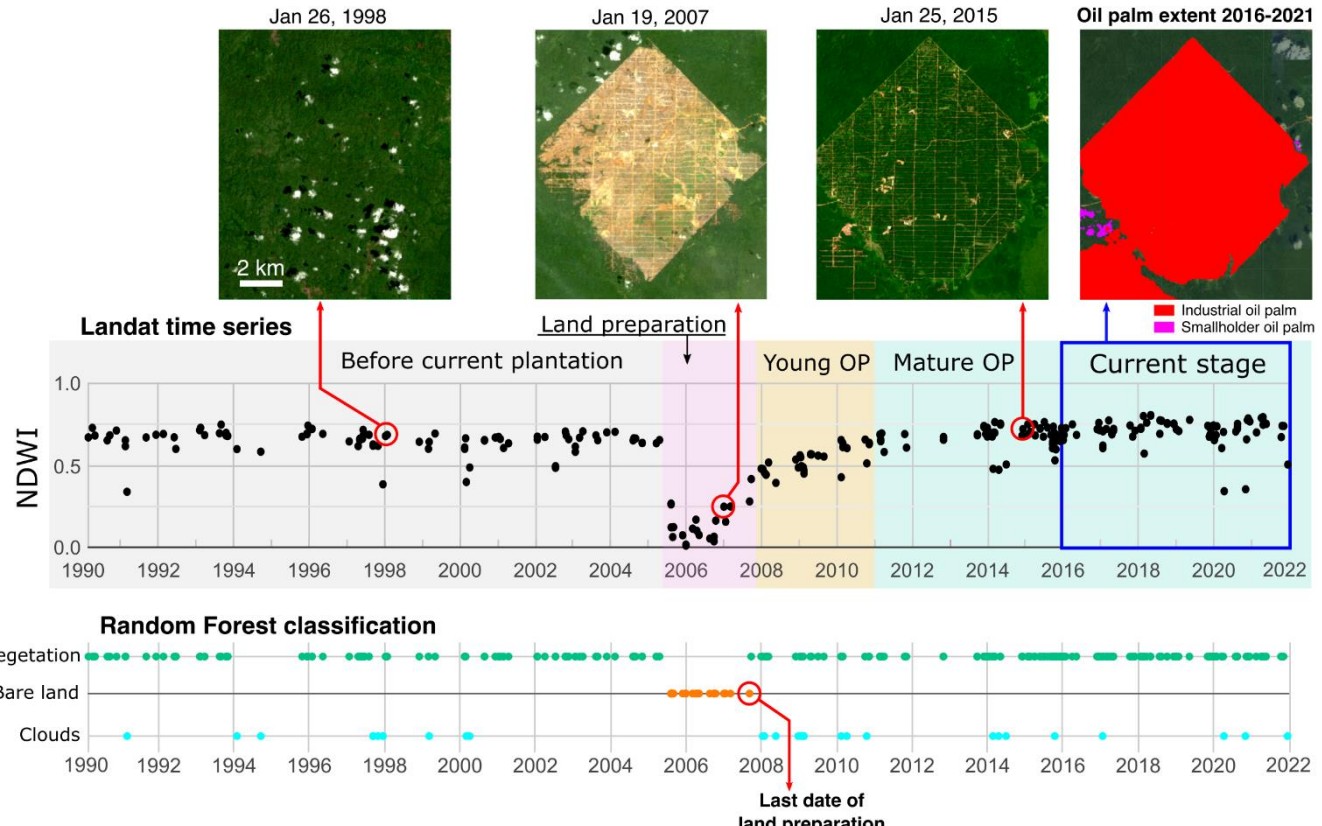

**Figure 2: Normalized difference water index (NDWI) and classification time series obtained from Landsat-5, -7, and -8 for an industrial oil palm plantation in South Papua, Indonesia. The time series was extracted for a pixel located at the center of the plantation. The upper images illustrate Landsat images for specific dates of the period 1990–2021. The current extent of oil palm was detected using Sentinel-1 data for the period 2016–2021. The Random Forest classified Landsat images into three classes: vegetation, bare land, and clouds.**

### 2.3.2 Estimation of the planting year

We estimated the timing of land development for oil palm using a Random Forest model that classified the Landsat time series into three classes (Fig. 2): vegetation, bare land, and clouds. The purpose of the Random Forest classification was to detect the last Landsat observations depicting bare land, which reflect the preparation of land for the current oil palm plantation, as of 2021. We included the class 'clouds' in the Random Forest to flag cloud observations that remained present in the Landsat images after applying the cloud masking. The Random Forest was applied only to the pixels in which oil palm was detected for the period 2016–2021. Then, we extracted the date of the Landsat observation corresponding to the final stage of land preparation. We named this date the last date of land preparation (LDLP), and it corresponds to the last Landsat observation that was classified as bare land. We assumed that the date of such Landsat observation indicates the time when the current plantation begins to grow and cover the ground, thereby altering the spectral signature of the pixel from bare land to vegetation.

We calculated the LDLP pixel-wise and applied a mode filter to reduce the noise and salt-and-pepper effect in the layer. The LDLP, however, is a proxy for the planting year and a mismatch may exist. To determine the planting year, we estimated it as the year with the lowest NDWI value observed in the five years preceding the LDLP. As an example, if the LDLP was detected in May 2016, we determined the planting year as the year with the lowest NDWI observed between June 2011 and May 2016. The agreement between the minimum NDWI year and the actual planting year was evaluated using field data (Subsection 2.3.5.2).

Our estimation reflects the planting year of the oil palm plantation detected between 2016 and 2021. Our definition of the planting year may not coincide with the year when oil palm was established for the first time, which could be substantially earlier than our estimate. This scenario arises in plantations that undergo a rotation, wherein oil palm is clear-cut and subsequently replanted with new oil palm seedlings.

## 2.3 Validation of the planting year

### 2.3.3.1 Visual inspection of the Landsat time series

We visually inspected the Landsat time series to verify that our methodology correctly detected the date when land was prepared for oil palm development. To do that, we plotted the NDWI time series, and an interpreter visualized the Landsat image that corresponded to the end of the land preparation for oil palm. The Landsat image that depicts the land preparation for oil palm is characterized by homogeneous bare land within the plantation area (Fig. A3a). The interpreter visualized the Landsat time series for all validation points that the deep learning model correctly classified as oil palm. Finally, we used a contingency table to compare the year determined by the interpreter with the LDLP image estimated using Landsat.

### 2.3.3.2 Comparison with field data

The accuracy of the planting year layer was assessed with 6,843 ground-truth observations of planting year obtained from plantation owners. This field data includes the planting year of 5,831 industrial parcels and 1,012 smallholder plantations, ranging from 1990 to 2021. A parcel is a subunit of an industrial plantation that encompasses any continuous oil palm area, usually delineated by the harvesting road network of the plantation. The industrial plantations are located in three countries— Brazil, Gabon, and Indonesia— and the ground-truth data was provided to us by the respective owning companies. The year of planting within smallholder plantations was acquired through interviews with the plantation owners. 93.5% of the interviews were carried out by the company that delivered the data for industrial plantations in Indonesia, and the remaining interviews were conducted specifically for the purpose of this study. The ground-truth data for the smallholder plantations were collected in Cameroon and Indonesia.

## 3 Results

### 3.1 Global oil palm extent

The oil palm extent layer is provided in 609 raster files covering an area of 100 x 100 km each. These 100 x 100 km grid cells represent the regions where oil palm was identified in the 2019 version, presented in Descals et al., 2021, as well as grid cells where oil palm was omitted in the previous version (Fig. A4). The new regions mainly include an oil palm hotspot in the state of Andhra Pradesh in India, industrial plantations in the Congo basin, and scattered plantations in Thailand and Central and South America. The classification model incorrectly identified some coconut plantations and paddy fields as oil palm, especially in Southeast Asia (Fig. A5). These false positives were also present in the previous 2019 oil palm layer but have been manually removed in this current study. We discarded a total of 82 grid cells that were processed in the 2019 oil palm layer, as these grid cells did not include any oil palm. These grid cells were predominantly located in coconut-growing regions in Mexico, India, the Philippines, and Indonesia.

We mapped a global oil palm area of 23.98 Mha, with 16.69 Mha (69.6%) corresponding to industrial plantations and 7.29 Mha (30.4%) to smallholders. 83.6% of the mapped area falls in Malaysia and Indonesia; Fig. 3 shows the global hotspot regions of oil palm cultivation. The mapped area is larger than the 2019 global oil palm layer because the updated version includes young oil palm, either from emerging plantations that reached the full canopy closure during the period 2016–2021 or from existing plantations that were replanted before 2019 (Fig. A6). The total oil palm area estimate is $16.82 \pm 0.19$ Mha for industrial plantations and $7.37 \pm 0.25$ Mha for smallholder plantations, with Indonesia being the top-producing country with $9.73 \pm 0.15$ Mha of industrial and $3.91 \pm 0.17$ Mha of smallholder plantations; a detail of the resulting layers for a hotspot area in Indonesia is shown in Fig. 4.

Our area estimates for Indonesia align with a previous study (Gaveau et al., 2022), which used a manually digitized reference dataset and reported 10.32 Mha of industrial and 5.92 Mha of smallholder oil palm for Indonesia in 2019. The higher oil palm area reported by Gaveau et al., 2022 includes land cleared for oil palm in 2019 that may not have matured into detectable oil palm in our classification map. This different definition of oil palm may explain the slightly higher oil palm area reported in Gaveau et al., 2022. Our area estimates also align with national statistics for oil palm harvested areas reported by FAO and USDA (Figure A7). The largest discrepancy occurred in Nigeria, where we estimated $0.38 \pm 0.13$ Mha, compared to the 4.86 Mha and 3.00 Mha reported by FAO and FAS-USDA, respectively. This difference may result from the inclusion of semi-wild oil palms in the FAO and USDA statistics. Semi-wild oil palm, common in West Africa, is mostly omitted in our oil palm layer as these palms typically grow scattered across the landscape, making them difficult to map accurately with Sentinel-1.

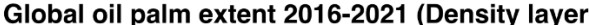

## Global oil palm extent 2016-2021 (Density layer)

**Figure 3: Global oil palm density map showing the density of oil palm at a 5 km resolution, derived from the 10 m global oil palm layer for the period 2016-2021.**

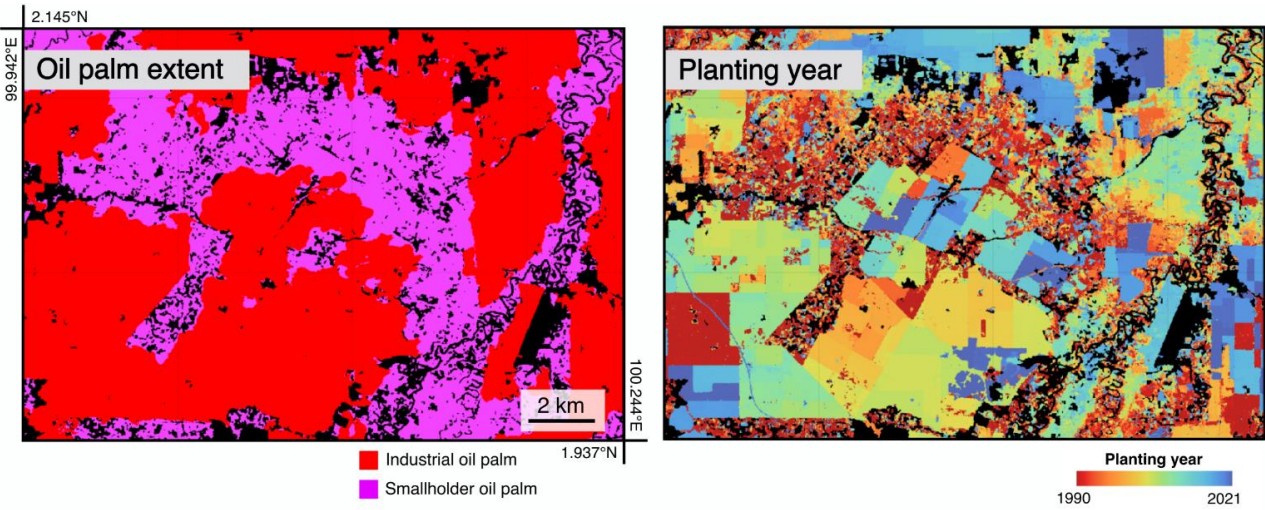

**Figure 4: Oil palm extent and planting year in a region in North Sumatra, Indonesia. The oil palm extent was obtained from the classification of annual Sentinel-1 composites for 2016–2021. The oil palm age was estimated from the Landsat time series for the period 1990–2021.**

## 3.2 Validation of the oil palm extent

The global oil palm extent 2016–2021 represents an improvement compared to the same layer developed for 2019. The overall accuracy was significantly higher in the updated oil palm layer (98.8 ± 0.1%; 95% confidence interval) compared to the accuracy of the 2019 layer (98.2 ± 0.1%) (Table 2). The producer's accuracy improved for all three classes. The producer's accuracy for industrial and smallholder oil palm increased from 74.6 ± 2.7% and 57.6 ± 1.9% to 91.0 ± 2.5% and 71.7 ± 1.2%, respectively, indicating that the updated layer misses far fewer oil palm plantations than the 2019 layer. The user's accuracy for smallholder oil palm also improved, from 67.3 ± 2.9% to 72.4 ± 1.8%, but decreased for industrial oil palm, from 94.5 ± 1.0% to 91.8 ± 0.7%, indicating that the updated version commits more industrial oil palm. This may be due to the rules used for merging the annual classification maps, which favored the presence of industrial oil palm. Despite the decrease in the producer's accuracy, the accuracies for industrial oil palm remain higher than those for smallholder plantations, indicating that mapping smallholder oil palm using satellite data remains challenging. In the oil palm extent 2016–2021, the producer's and user's accuracies are of the same order for industrial and smallholder oil palm, meaning that the model omits and commits roughly the same area for both classes. This explains the small difference between the area estimate and the area mapped in this study.

**Table 2. Accuracy assessment of the global oil palm layer (Descals et. al, 2021) and the oil palm extent 2016-2021 presented in this study. The accuracy metrics are the Overall Accuracy (OA), the user's accuracy (UA), and the producer's accuracy (PA). Upper and lower bound estimates for a 95% confidence level are shown in parenthesis.**

| | | Global oil palm layer 2019 (Descals et. al, 2021) | Oil palm extent 2016-2021 (this study) |
|---|---|---|---|
| OA (%) | | 98.2 (98.1, 98.2) | 98.8 (98.8, 98.9) |
| | Other | 98.7 (98.6, 98.8) | 99.6 (99.5, 99.6) |
| | Industrial oil palm | 94.5 (93.5, 95.4) | 91.8 (91.0, 92.5) |
| UA (%) | Smallholder oil palm | 67.3 (64.4, 70.2) | 72.4 (70.6, 74.2) |
| | Other | 99.8 (99.7, 99.8) | 99.6 (99.6, 99.7) |
| | Industrial oil palm | 74.6 (71.9, 77.3) | 91.0 (88.5, 93.5) |
| PA (%) | Smallholder oil palm | 57.6 (55.7, 59.5) | 71.7 (70.5, 72.9) |
| Mapped industrial oil palm (Mha) | | 13.10 | 16.69 |
| Mapped smallholder oil palm (Mha) | | 6.37 | 7.29 |
| Area estimate industrial oil palm (Mha) | | 16.59 (16.25, 16.92) | 16.82 (16.63, 17.01) |
| Area estimate smallholder oil palm (Mha) | | 7.45 (7.11, 7.78) | 7.37 (7.12, 7.62) |

### 3.3 Validation of the planting year

The LDLP agreed with the true date obtained by visual inspection of the Landsat time series by 59.6%; 501 points out of the 841 points (Fig. 5). This agreement was 64.8% (405/625) for industrial and 44.0% (95/216) for smallholder plantations. 76.3% of the points showed an error of ±3 years: 78.4% for industrial and 61.6% for smallholders. Many of these validation points were located near roads or the border of a plantation, where the reliability of the LDLP may be lower due to the border effect. The border effect refers to the high heterogeneity in land cover types observed within a 30-meter Landsat pixel at the edge of the plantation. This variation in land cover can obscure the changes in the plantation reflected in the NDWI time series. In smallholder plantations, there are more pixels that are affected by the border effect, and validation points are more likely to be at the edges than in industrial plantations. This may explain the lower agreement in smallholder plantations. Additionally, smallholder plantations are often established over several years and a question from our field teams about the initial year of establishment may have been confusing to the plantation owner (see Discussion).

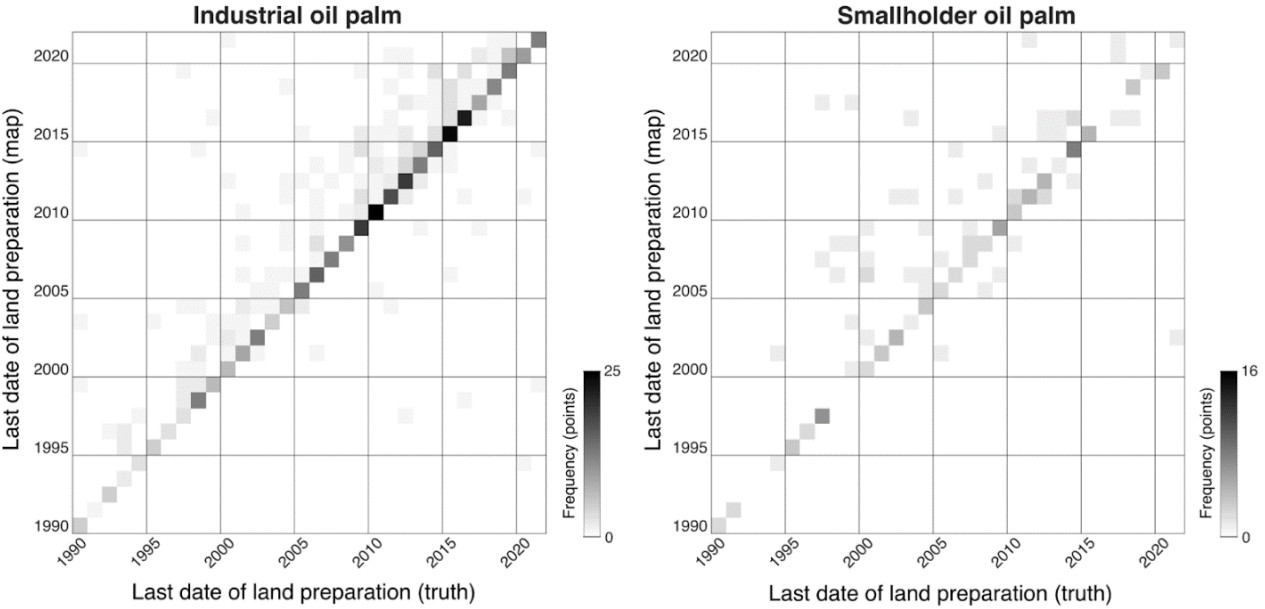

**Figure 5: Contingency table between the 'last date of land preparation' estimated from Landsat using a Random Forest classification (map) and the same date obtained by visual inspection of the Landsat time series (truth). The contingency table was created with 625 validation points in industrial oil palm plantations and 216 points in smallholder plantations, respectively.**

The agreement between LDLP and the true date obtained by visual inspection varied by region. We found a low agreement in regions where the Landsat NDWI time series presents seasonal fluctuations. In these regions, NDWI decreases during annually recurring dry periods, and the Random Forest incorrectly identified these low NDWI values as bare land. We identified this issue in India and various parts of Southeast Asia, including Thailand and Sulawesi (Fig. A3b). The agreement was also low

in Africa, mostly explained by the high frequency of clouds and gaps in the Landsat time series, which affected the period 1990-2014 (Fig. A3c).

The comparison with ground-truth data showed a good agreement (Fig. 6), with an overall mean error (ME; field data - satellite estimate) of -0.24 years, a root-mean-squared error (RMSE) of 2.65 years, and a coefficient of correlation ($R^2$) of 0.86. The agreement in terms of RMSE was higher for industrial plantations (2.02 years) than for smallholder plantations (4.89 years). The high agreement in industrial plantations was found across all regions (Table 3), indicating that the planting year can be determined accurately for industrial oil palm. The difference between the field data and the satellite estimation was consistent

across the range of planting years, although slightly higher for older plantations. Smallholders in Cameroon showed the lowest agreement (RMSE = 7.01 years).

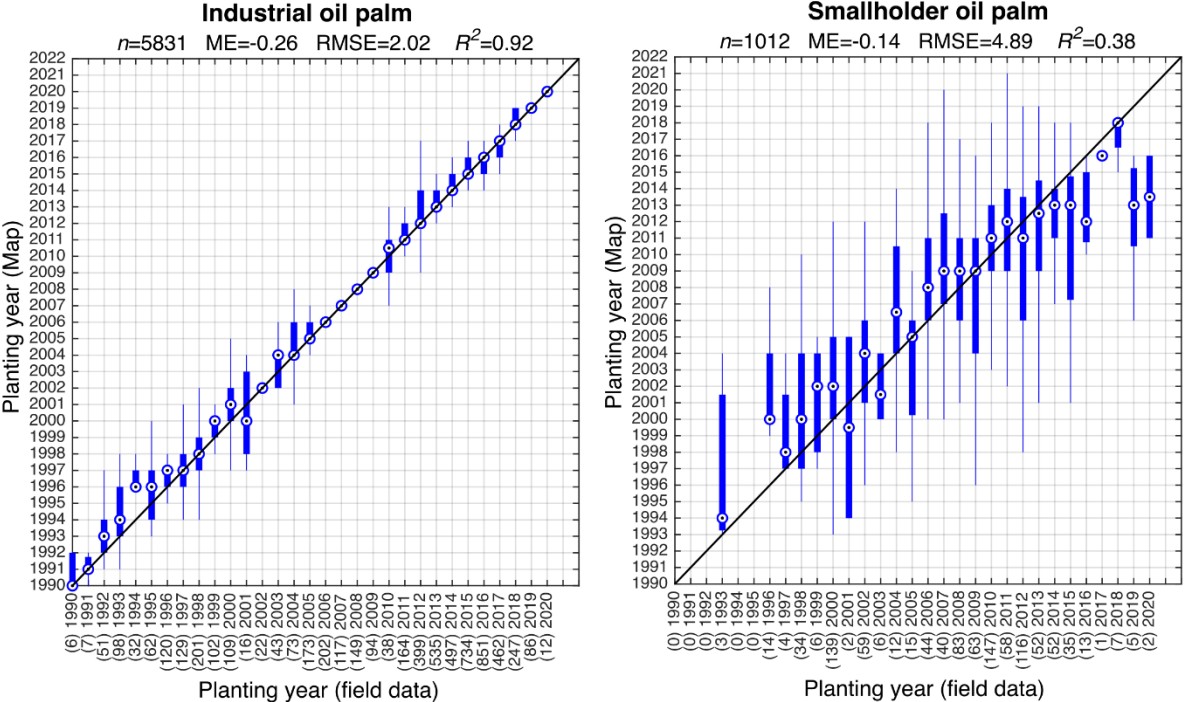

**Figure 6: Comparison between the planting year obtained from Landsat and the true planting year obtained from field data in a) industrial and b) smallholder oil palm plantations. The title shows the number of observations (_n_), mean error (ME; field data - satellite estimate), root-mean-squared error (RMSE), and coefficient of correlation (_R²_). The number of observations from field data for each year appears in parenthesis in the x-axis.**

The dynamic of planting years follows a similar pattern for industrial and smallholder plantations in Indonesia, Malaysia, and around the world (Fig. 7). Since 1990, the area of oil palm planted each year increased steadily, peaking around 2015. However,

after 2015, this trend reversed and the area of oil palm planted each year declined.. This indicates that most oil palms are of medium age; the average age of industrial oil palm is 14.0 years, while smallholders have an average of 14.2 years. The area of oil palm over 20 years is 4.34 Mha for industrial oil palm and 1.94 Mha for smallholders.

Table 3: Comparison between the planting year estimated from Landsat and the true planting year obtained from field data for different regions. The columns show the mean error (ME), root-mean-squared error (RMSE), coefficient of correlation ($R^2$), and the number of observations ($n$).

| Region | Plantation type | ME (years) | RMSE (years) | $R^2$ | $n$ |
|---|---|---|---|---|---|
| ALL | All | -0.24 | 2.65 | 0.86 | 6843 |
| ALL industrial | Industrial | -0.26 | 2.02 | 0.92 | 5831 |
| ALL smallholders | Smallholder | -0.14 | 4.89 | 0.38 | 1012 |
| Brazil | Industrial | 0.44 | 1.70 | 0.97 | 64 |
| Liberia | Industrial | -1.45 | 1.91 | 0.18 | 179 |
| Gabon | Industrial | 0.04 | 2.38 | 0.26 | 2511 |
| North Sumatra | Industrial | -0.32 | 1.14 | 0.98 | 664 |
| Riau | Industrial | -0.36 | 1.53 | 0.97 | 116 |
| Bangka-Belitung | Industrial | -0.54 | 1.82 | 0.97 | 610 |
| West Kalimantan | Industrial | -0.59 | 1.88 | 0.87 | 658 |
| Central Kalimantan | Industrial | -0.66 | 2.33 | 0.86 | 224 |
| East Kalimantan | Industrial | -0.79 | 2.18 | 0.84 | 358 |
| West Papua | Industrial | 0.07 | 0.88 | 0.73 | 447 |
| Cameroon | Smallholder | -2.36 | 7.01 | 0.19 | 11 |
| North Sumatra | Smallholder | 1.06 | 5.55 | 0.29 | 581 |
| Riau | Smallholder | -2.83 | 4.45 | 0.41 | 58 |
| Bangka-Belitung | Smallholder | -1.61 | 3.52 | 0.38 | 307 |
| West Kalimantan | Smallholder | -1.40 | 4.00 | 0.48 | 55 |

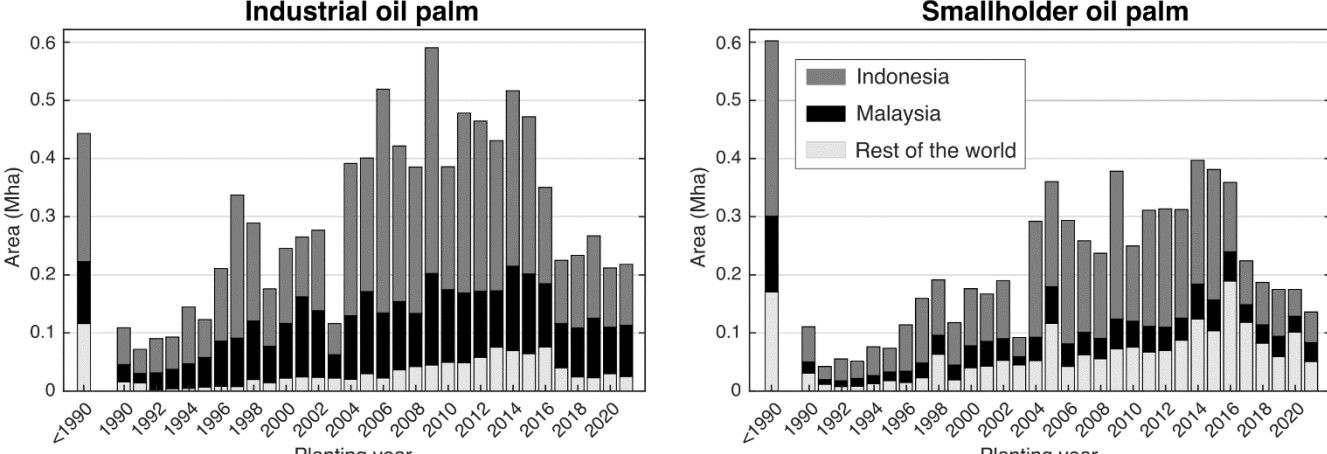

Figure 7: Oil palm planting year in industrial and smallholder oil palm for the period 1990–2021 in Indonesia, Malaysia, and the rest of the world. The planting year denotes the time when oil palm was established, either as a new plantation or through the process of replanting, where old oil palm trees are cleared and replaced by young ones to maintain productivity.

## 4 Discussion

This study presents a dataset that includes a global oil palm extent layer for 2021 and an oil palm planting year layer from 1990 to 2021. The oil palm extent layer improves the omission error in the previously published map for 2019 (Descals et al., 2021). Two factors explain this improvement in the omission error. First, the current methodology can identify young oil palm plantations that have been replanted during the period 2016–2021. Second, by combining multiple annual classifications, we minimized the omission of oil palm. Oil palm can be excluded in a single year, potentially due to inherent noise and the impact of surface roughness in radar Sentinel-1 data. The merging of annual classifications favored the classification of oil palm, thus reducing the omission of oil palm. The updated oil palm layer also improves in terms of commission error; false positives in coconut, paddy fields, and other palm species were removed manually. Despite the improved accuracies, the oil palm extent layer presents three major caveats. First, the oil palm extent layer mostly omits emerging young oil palm that were planted for the first time in 2016–2021. Our layer particularly omits plantations established in the later years of the period 2016–2021, as these new plantations had not yet become detectable by Sentinel-1 by 2021. This is because young oil palm exhibits a spectral backscatter in Sentinel-1 that is similar to other non-oil palm land covers. Second, we assumed that oil palm plantations that were clearcut between 2016 and 2021 are replanted as oil palm. Oil palm plantations that undergo replanting with a different tree plantation or crop would be regarded as false positives. Third, most open-canopy oil palm remains undetected in this updated version, although this issue was more apparent in the 2019 oil palm layer presented in Descals et al., 2021. This affects failed plantations through burning, floods, and pests (Fosch et al., 2023), as well as semi-wild plantations and oil palms in heterogeneous settings, especially in Africa. Subsistence-level palm oil in Africa could add millions of hectares; areas of these unaccounted traditional oil palm plantations were estimated to be 6.66 Mha in Africa in 2013 (Carrere, 2010). The presence of unaccounted semi-wild oil palms likely explains the ~4.5 Mha discrepancy between our area estimates and FAO's oil palm area in Nigeria, as well as the difference between our global oil palm mapped area (23.98 Mha) and the FAO's reported global harvested area (29.62 Mha) for 2021. Despite this discrepancy, the comparison with official statistics supports the validity of our oil palm extent layer, as our area estimates closely align with the FAO and USDA-reported oil palm areas in other top-producing countries.

The validation dataset provided in this study is also an improvement over the previous version published in Descals et al., 2021. First, we corrected the true label in the validation points that fell in coconut plantations and were incorrectly interpreted as oil palm. Young oil palm plantations in 2019, which were labeled as class 'other', were also corrected and re-labeled as oil palm. The changes in the validation dataset explain the lower producer's and user's accuracy obtained in this study for the 2019 oil palm layer compared to the same accuracy presented in Descals et al., 2021. Second, we provide information whether the validation points were generated by simple random sampling or stratified random sampling. Simple random sampling is ideal for assessing and comparing the accuracy of different oil palm datasets, as it eliminates the need to recalculate the number of points in each stratum, unlike stratified random sampling (Gaveau et al., 2021). Future research can directly use our

validation dataset to assess the accuracy of oil palm datasets, creating statistically rigorous accuracy metrics and area estimates along with confidence intervals (Sheil et al., 2024).

The accuracy of the planting year layer was assessed using a large ground-truth dataset. The comparison with ground-truth data allowed us to determine which Landsat observation matched most closely with the true year of planting. In this study, we found that the minimum NDWI date showed high agreement with the planting year from field data, particularly for industrial oil palm. These are important findings that should be considered when estimating oil palm development in future studies. While the agreement, in terms of RMSE, was suboptimal for smallholders, the bias was negligible. The low agreement for smallholders might be partially explained by a memory or reporting error; the longer ago a plantation was developed, the less likely it is that the year of planting was reported accurately. Furthermore, smallholder plantations often develop sequentially, beginning with a few hectares and expanding in subsequent years, making it challenging to answer interview questions about the planting year. Additional limitations could explain the inaccurate estimations of the planting year for smallholders. First, the border effect might obscure the land cover changes reflected in the NDWI time series, especially in small plantations where the 30-meter Landsat pixels do not cover the entire plantation. Second, our methodology cannot detect replanting years in cases where owners use the underplanting technique, which involves planting seedlings between older trees and then removing the older trees once the seedlings have matured into young trees (Chia et al., 2002). This replanting approach would primarily concern smallholders, as industrial plantations typically clear-cut existing oil palm trees and replant entire areas with new palms afterwards. Lastly, we observed inaccurate estimations of the planting year in areas where the Landsat NDWI time series exhibited significant gaps due to data scarcity or extensive cloud cover. These data gaps are especially pronounced in the oil palm-growing regions of Africa (Kovalskyy and Roy, 2013), in particular before 2013, when only Landsat-5 and -7 data were available. Inaccuracies were also found in regions where vegetation displayed seasonal fluctuations in NDWI. These issues were most prevalent across much of Africa, India, and specific regions of Southeast Asia. Users should exercise caution when using our year of planting layer, particularly in these areas and for smallholders in general.

The trends in the planting year show patterns that are comparable to a previous study (Gaveau et al., 2022), which presented a recent decline in oil palm expansion in Indonesia from around 2010. Despite the similarities, our trends are not entirely comparable to those obtained by Gaveau et al., 2022. Their analysis measures the number of new plantation areas (expansion) added each year, whereas our annual oil palm area encompasses both annual expansion and the clearing and replanting of old plantations. In addition, our methodology presents a caveat; our classification model only detects closed-canopy plantations, thereby excluding very young plantations with open canopies. An oil palm plantation typically takes about 3 years or longer to achieve full canopy closure, at which point radar satellites can detect it. As a result, using imagery up to 2021, we could not detect new plantation areas established during the last three years of the time series (2019–2021). This limitation implies that the oil palm area for planting years after 2019 mostly includes replanting of existing old plantations and overlooks the expansion of new areas of land.

Our planting year layer provides insight into where and how much area requires replanting. Replanting costs are estimated to be between 3,200 and 3,800 euros per hectare (Nurfatriani et al., 2019). These costs are important, especially in Indonesia, where replanting will soon be necessary in many oil palm plantations that have surpassed their ideal production age (Grass et al., 2020). According to our results, Indonesia has 3.54 Mha of oil palm older than 20 years, indicating a substantial capital expenditure of some USD 10 to 15 billion over the decade after 2021. 30.5% of this area that requires replanting is on smallholder plantations, where the availability of funds to finance replanting may be a significant constraint (Petri et al., 2023). Replanting is important because it allows higher-yielding varieties to replace older lower-yield ones (Ismail and Mamat, 2002), and, given the growing demand for vegetable oils (Meijaard et al., 2024), effective replanting can reduce the expansion of the oil palm area or other oil crops and, thus, minimize negative environmental and social impacts. Replanting also represents an opportunity to develop more wildlife friendly plantations, by establishing forest corridors (Gregory et al., 2014). Finally, the methodology we develop for mapping oil palm age can be relevant for regulations such as the European Commission's "European Union Deforestation Free-Product Regulation" (EUDR). EUDR has already entered into force, which bans certain commodities coming from areas deforested and planted with those selected commodities after December 31[st], 2020, from entering the EU market. Our methodology can be adjusted to annually estimate the timing of establishment for emerging oil palm plantations that result from recent deforestation, enabling the monitoring and reporting of deforestation driven by oil palm expansion.

## 5 Code availability

The code that generates the Sentinel-1 and Sentinel-2 composites can be found at: https://doi.org/10.5281/zenodo.4617748 (Descals, 2021).

The original code of the semantic segmentation model DeepLabv3+ can be found at: https://github.com/tensorflow/models/tree/master/research/deeplab (GitHub, 2021).

## 6 Data availability

The dataset presented in this study is freely available for download at https://doi.org/10.5281/zenodo.13379129 (Descals, 2024). The repository in Zenodo contains the following data:

- Grid_OilPalm2016-2021.shp: shapefile that delineates the 609 grid cells of 100 x 100 km where oil palm was found.
- GlobalOilPalm_OP-extent.zip: 609 raster tiles of 100 x 100 km in geotiff format. The raster files show the results of a deep learning classification of Sentinel-1 data at a spatial resolution of 10 meters. The classes are the following:

    [0] Other land covers that are not oil palm.

[1] Industrial oil palm plantations

450         [2] Smallholder oil palm plantations.

- GlobalOilPalm_YoP.zip: 609 raster tiles of 100 x 100 km in geotiff format. The raster files depict the year of oil palm plantation. The raster files have a spatial resolution of 30 meters.

- Validation_points_GlobalOP2016-2021.shp: shapefile that contains the 18,812 points used to validate the global oil palm extent 2016–2021 and the oil palm age layer. Each point includes the attribute 'Class', which is the class assigned by visual

interpretation of sub-meter resolution images, and the attributes 'OP2016-2021' and 'OP2019', which show the mapped classes in the oil palm extent 2016–2021 (this dataset) and the global oil palm layer 2019 (Descals et al., 2021), respectively. These attributes contain the following class values:

        [0] Other land covers that are not oil palm.

        [1] Industrial oil palm plantations.

460         [2] Smallholder oil palm plantations.

The Sentinel-1 SAR GRD is available at the Copernicus Open Access Hub: https://scihub.copernicus.eu/ (last access: 25 April 2024). The Landsat-5, -7, and -8 surface reflectances are available at the USGS Earth Explorer portal: https://earthexplorer.usgs.gov/ (last access: 25 April 2024). Very high-resolution images (spatial resolution <1 m) can be

visualized in the Google Earth Engine code editor or Google Maps. We obtained the 2021 official oil palm area statistics from FAOSTAT (FAO, 2022) and FAS-USDA (FAS-USDA, 2024).

The oil palm extent and the planting year can be visualized at: https://ee-globaloilpalm.projects.earthengine.app/view/global-oil-palm-planting-year-1990-2021 (last access: 25 April 2024). This web map allows for the inspection of Landsat time series

and the visualization of historical satellite images for a given oil palm plantation.

## 7 Conclusions

This study offers significant advances by providing global layers on oil palm extent and planting year, which are critical for understanding the environmental impacts associated with oil palm. Our methodology improves the omission error by detecting oil palm rotations and reducing false positives through manual removal of misclassified areas, resulting in a mapped global oil

palm area of 23.98 Mha. However, caveats exist, including the omission of young plantations, assumptions about replanting, and challenges in detecting open-canopy oil palm. The validation dataset can help future studies in evaluating upcoming global oil palm datasets. Finally, our estimation of oil palm age provides insights into replanting needs, which is crucial for sustainable management and addressing environmental concerns.

## 8 Author contributions

The conceptualization for this work originated from AD and EM. AD designed the study. AD collected the reference points, implemented the data processing workflow, and generated the figures and tables. AD and EM wrote the draft and AD, DG, SW, ZS, and EM were involved in the revision of the manuscript.

## 9 Competing interests

Adrià Descals performed remote sensing consulting work for the ICT & GIS department at PT Austindo Nusantara Jaya Tbk.
David Gaveau is a member of the IUCN Oil Crops Task Force, a group tasked by the IUCN to investigate the sustainability of palm oil and he has done oil palm-related work for this task force, and Greenpeace. Serge Wich received research funding from PT Austindo Nusantara Jaya Tbk and is a member of the IUCN Oil Crops Task Force and he has done oil palm related work for this task force. Erik Meijaard chairs and has received funding from the IUCN Oil Crops Task Force and he has done work paid by palm oil companies and the Roundtable on Sustainable Palm Oil.

## 10 Financial support

We acknowledge funding from the Roundtable on Sustainable Palm Oil (RSPO) (funding agreement PUR-02750).

## 11 Acknowledgements

We thank Austindo Nusantara Jaya, OLAM Gabon, and Agropalma for providing age verification data and Andi Erman and Ayompe Lacour for coordinating field data collection.

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

**APPENDIX A**

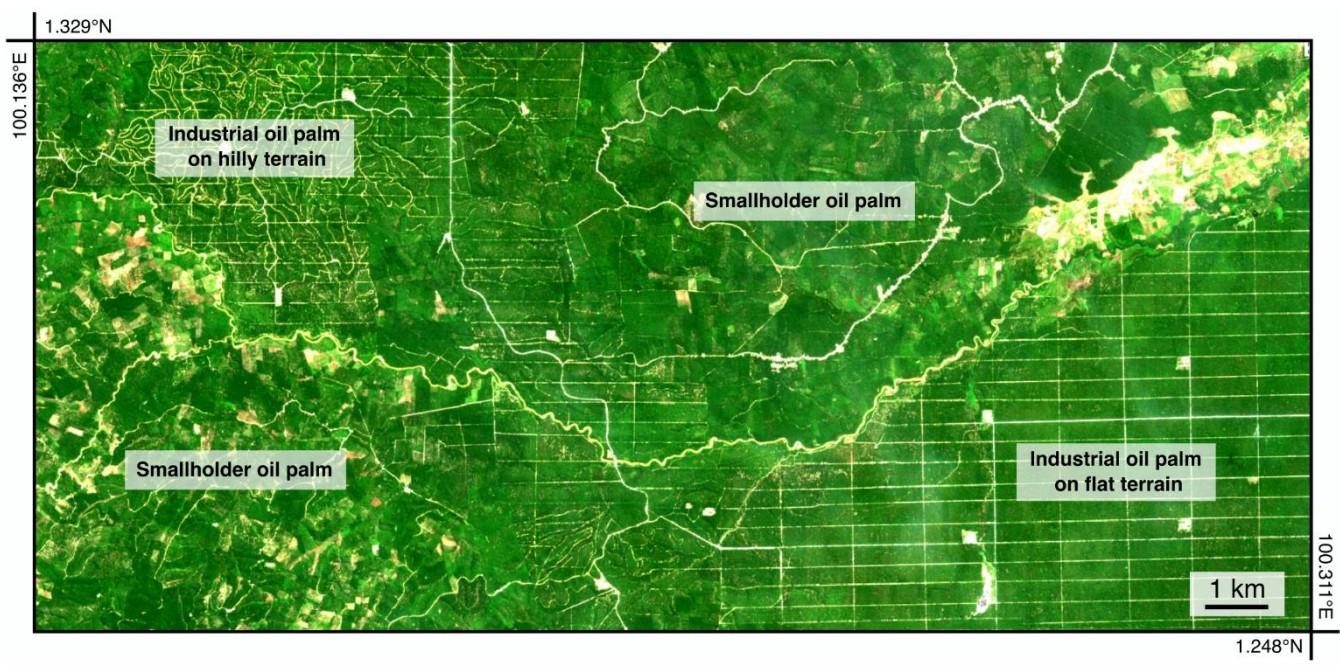

**Figure A1: Sentinel-2 true color composite depicting industrial and smallholder plantations in a region in Riau province, Indonesia.**

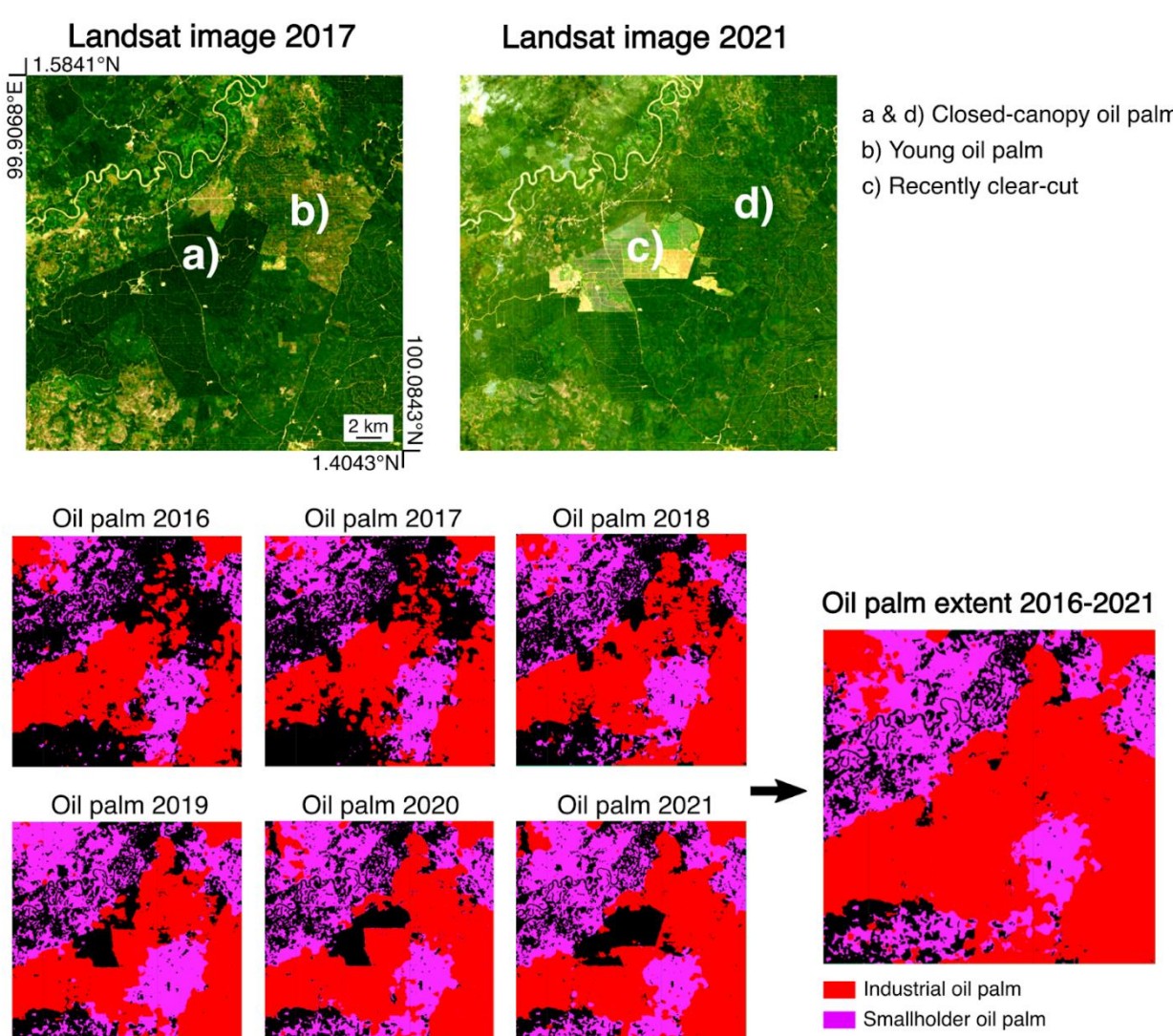

575

**Figure A2: Example of the merging of the multi-annual oil palm classification maps. The upper images show two Landsat images taken in 2016 and 2021 in a region in North Sumatra. The two images depict a clear-cutting of an industrial plantation — a) to c), and the transition of a young oil palm to a closed-canopy plantation — b) to d). The lower images show the annual oil palm classification maps from 2016 to 2021 and the merged oil palm layer in a region in North Sumatra. Young oil palm and oil palm from replanting were not detected in certain years of the annual classification maps, but these plantations were included in the merged layer 2016–2021.**

580

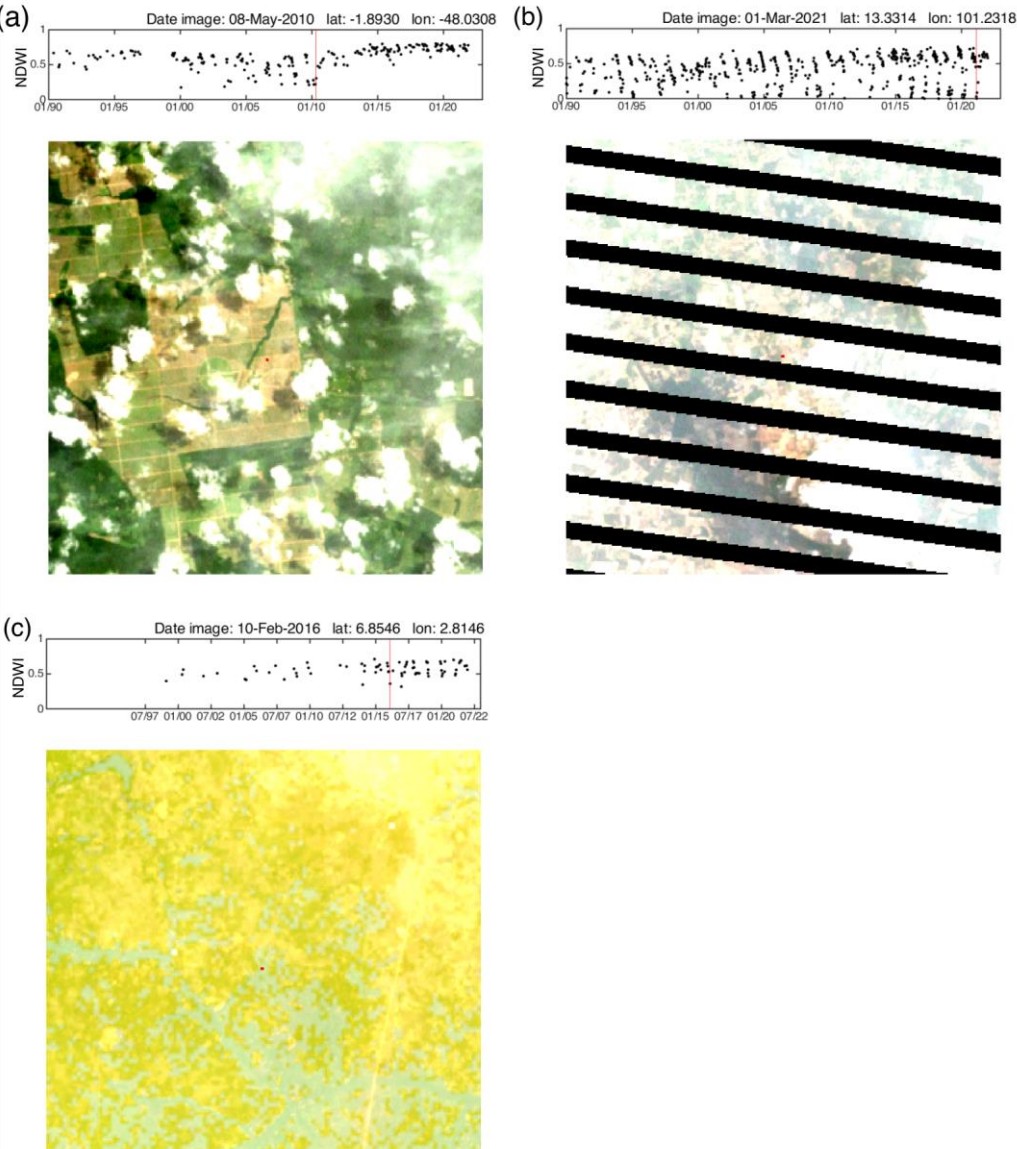

**Figure A3: Example of the Landsat NDWI time series and the Landsat image that corresponds to the last date of land preparation for three validation points. The NDWI time series corresponds to the pixel located at the center of the image. The images correspond to the last Landsat images classified as bare land using a Random Forest classification. (a) The NDWI time series shows the typical patterns of oil palm development, and the Landsat image corresponds to the land preparation for oil palm. (b) The NDWI time series presents a pronounced seasonality, and the Random Forest classified the low NDWI values as bare land. (c) The NDWI time series presents persistent gaps, and the last observation classified as bare land corresponds to an image largely impacted by cloud cover.**

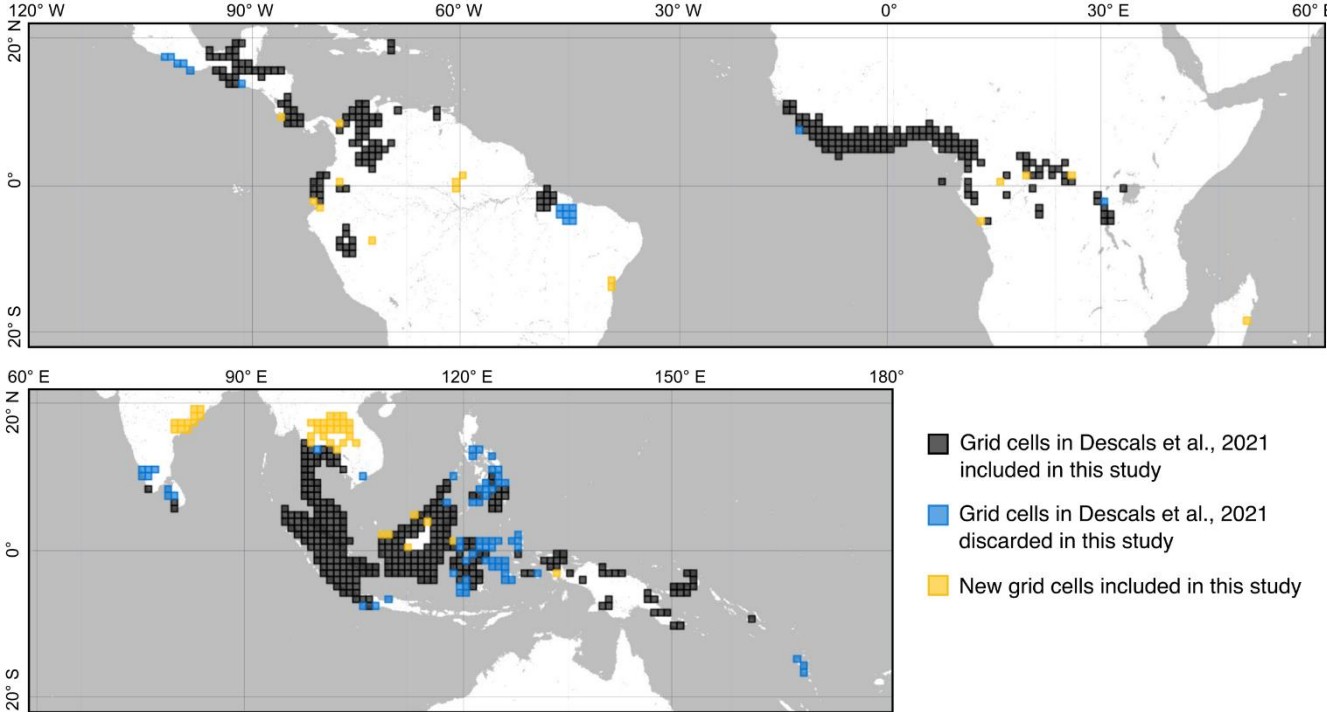

**Figure A4: Grid cells of 100 x 100 kilometers used for the classification of Sentinel-1 composites. Grid cells in black and blue show the regions where oil palm was detected for 2019 in a previous study (Descals et. al, 2021). Blue grid cells were not included in this study as they only included oil palm false positives. Grid cells in orange represent regions where oil palm was detected for the first time in this study.**

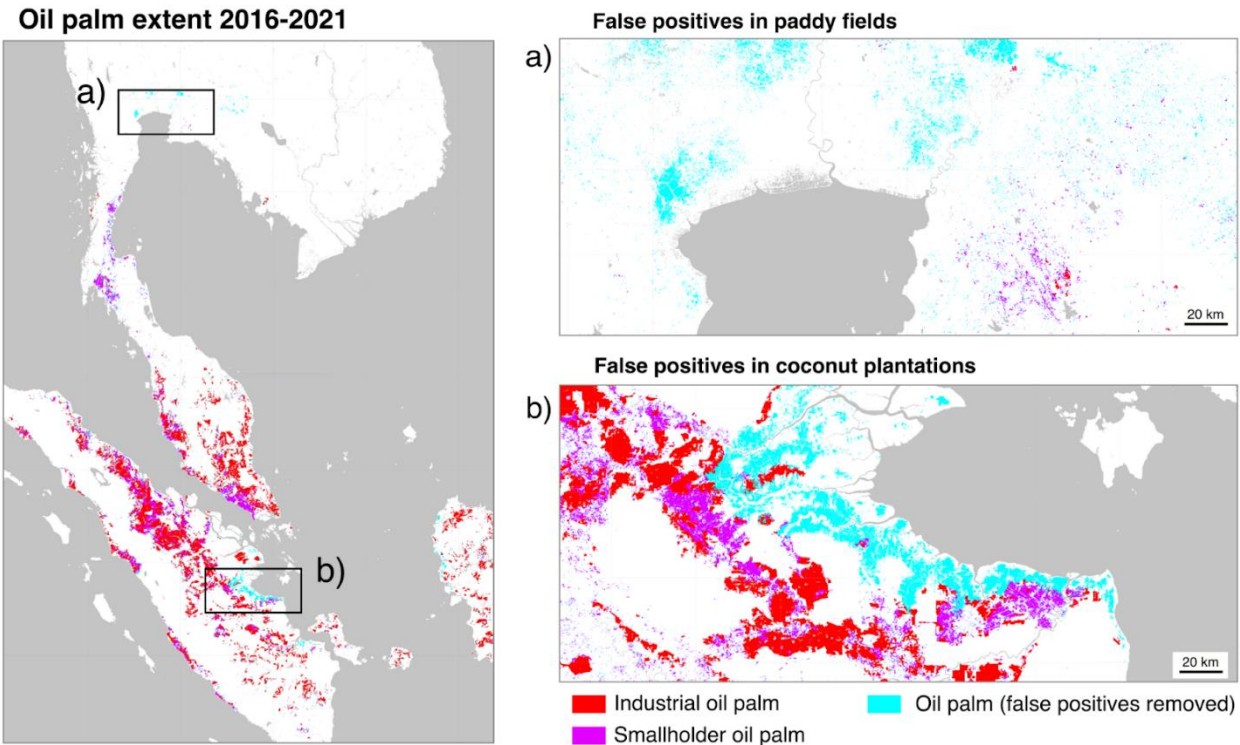

**Figure A5: Oil palm false positives in the oil palm extent 2016–2021. False positives (in cyan) were found in paddy fields (inset a) and coconut plantations (inset b) and were removed in the final version of the layer.**

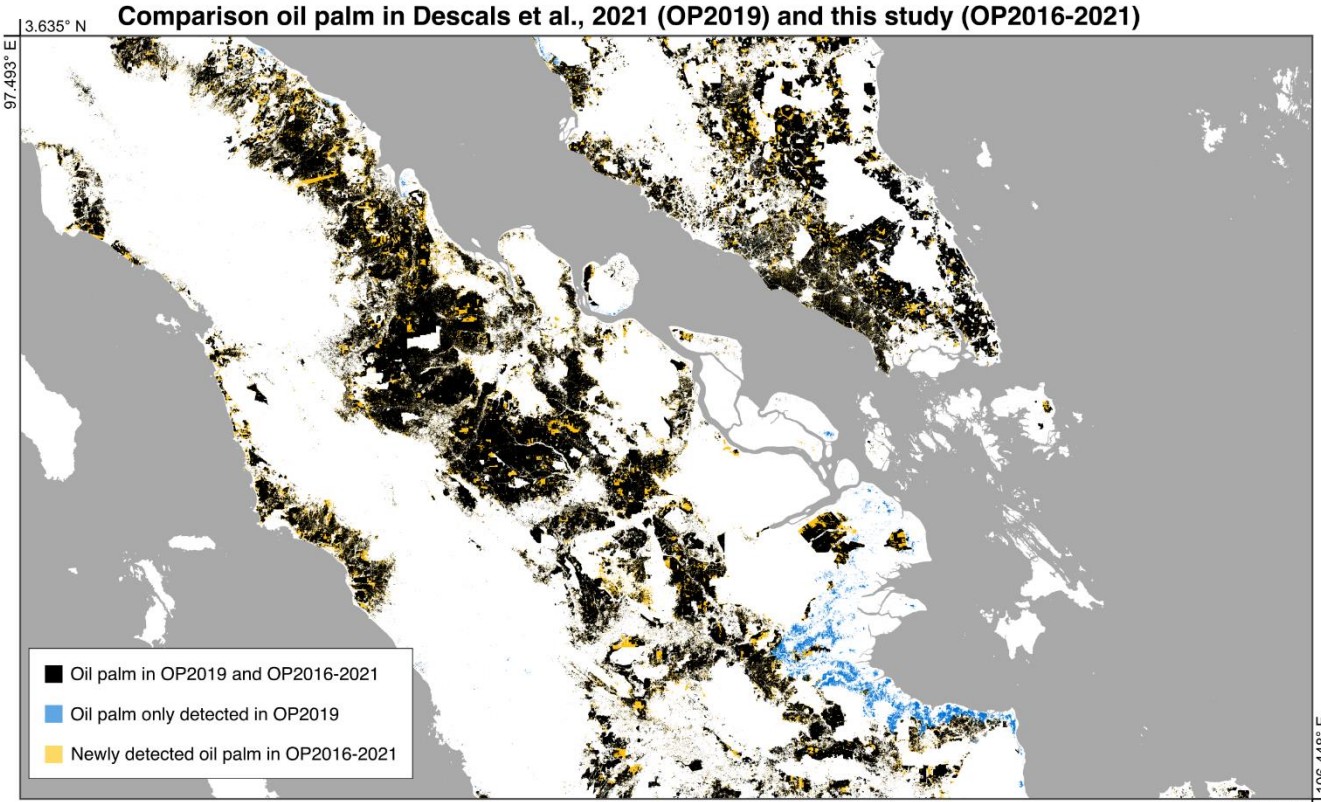

**Figure A6: Comparison between the oil palm dataset developed by Descals et al. (2021) for 2019 and the oil palm extent layer developed in this study for 2016-2021. Black pixels indicate oil palm detected in both datasets, orange pixels represent newly detected oil palm in this study, and blue pixels correspond to areas detected in 2019 but not included in this study. The orange pixels primarily correspond to young oil palm omitted in the previous version, while the blue pixels mostly consist of false positives in coconut-growing regions.**

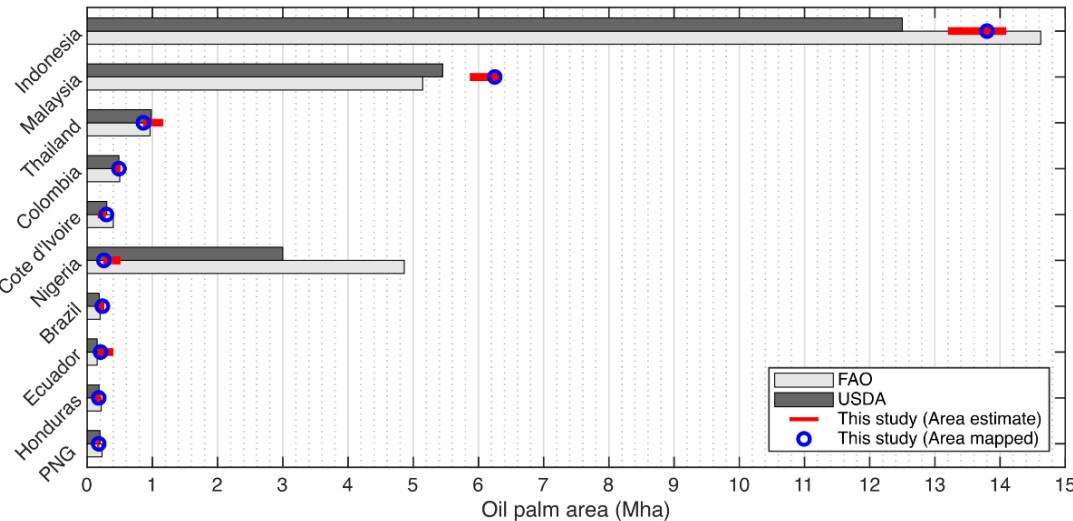

**Figure A7: Oil palm area for the 10 highest producing countries according to the dataset presented in this study. The bars depict the oil palm area for 2021 according to official statistics (FAO and USDA), the blue circles represent the mapped oil palm area using the deep learning model, and the red line shows our oil palm area estimate with a 95% confidence interval.**
