# Peer review of "Global mapping of oil palm planting year from 1990 to 2021"

_Earth System Science Data, 2024_

## Author Comment (AC1)

A very useful study but the 'headline' figure of OP global OP plantation area, which I assume is dated as of 2021 is given as 23.98 Mha.

However, just a few lines later an FAO estimate from 2022 is quoted as a much higher area of 30 Mha - this discrepancy is not discussed anywhere in the paper. A figure of 30 Mha is also reported by FAO in 2023 and is widely used by other sources such as https://ourworldindata.org/palm-oil

Moreover, the 2024 value quoted by USDA (which might just be using the FAO data) is also 30 Mha.

In contrast, the much-quoted 2022 value of 23 Mha by Statista is much closer to the current authors' value

It would be nice to have some discussion about these discrepancies, which are from reputable bodies like FAO, USDA etc, because these headline values are widely quoted and used by policymakers, NGOs, the media etc.

Response #1: Thank you for pointing out this issue in the manuscript. We have now included a new figure comparing our oil palm area estimates with the FAO and USDA statistics for the top 10 countries with the largest oil palm areas. Our estimates align well with the reported areas from both agencies, except for Nigeria. In Nigeria, our estimates differ from the FAO's by approximately 4.5 Mha, which largely accounts for the gap between our global oil palm mapped area (23.98 Mha) and the FAO's reported figure (29.62 Mha). The revised manuscript presents these results and discusses the likely reasons for this discrepancy in Nigeria (line 271).

"*Our area estimates also align with national statistics for oil palm harvested areas reported by FAO and USDA (Figure A7). The largest discrepancy occurred in Nigeria, where we estimated 0.38 ± 0.13 Mha, compared to the 4.86 Mha and 3.00 Mha reported by FAO and FAS-USDA, respectively. This difference may result from the inclusion of semi-wild oil palms in the FAO and USDA statistics. Semi-wild oil palm, common in West Africa, is mostly omitted in our oil palm layer as these palms typically grow scattered across the landscape, making them difficult to map accurately with Sentinel-1.*"

The revised manuscript also discusses the likely reason for the discrepancy between our total estimate and the global statistic from FAO (line 373):

"*Subsistence-level palm oil in Africa could add millions of hectares; areas of these unaccounted traditional oil palm plantations were estimated to be 6.66 Mha in Africa in 2013 (Carrere, 2010). The presence of unaccounted semi-wild oil palms likely explains the ~4.5 Mha discrepancy between our area estimates and FAO's oil palm area in Nigeria, as well as the difference between our global oil palm mapped area (23.98 Mha) and the FAO's reported global harvested area (29.62 Mha) for 2021. Despite this discrepancy, the comparison with official statistics supports the validity of our oil palm extent layer, as our area estimates closely align with the FAO and USDA-reported oil palm areas in other top-producing countries.*"

[Figure]

*Figure A7: Oil palm area for the 10 highest producing countries according to the dataset presented in this study. The bars depict the oil palm area for 2021 according to official statistics (FAO and USDA), the blue circles represent the mapped oil palm area using the deep learning model, and the red line shows our oil palm area estimate with a 95% confidence interval.*

We also extracted the oil palm area from the SPAM dataset, which is an average for 2019-2021. SPAM reports a total area of 28.6 Mha, identical to the FAO's reported area for 2020. This is because SPAM derives its crop maps from FAO data. Additionally, we retrieved the oil palm area from STATISTA for 2020, which reported 28.7 Mha—essentially the same as the FAO and SPAM figure for 2020. Since both SPAM and STATISTA likely rely on FAO, we did not include these estimates in the manuscript to avoid redundancy with FAO's data.

---

## Author Comment (AC2)

**Response to Referee #1**

Oil palm is an important oilseed that has received widespread attention because of its involvement in issues such as tropical deforestation. In this paper, based on their 2021 global oil palm mapping, the algorithm is optimized to achieve a higher accuracy of the global 10m oil palm mapping. Meanwhile, they further track the planting year of oil palms from 1990 to 2021 and demonstrated to achieve the fulfilling performance. In summary, this manuscript is well-written and presents a high-quality global oil palm dataset, which meets the ESSD standard for high data quality. Below are some comments to this manuscript:

Response #1: Thank you for your positive remarks about the quality of the dataset. We have revised the manuscript and made the necessary corrections. Please see below a detailed response to all the comments.

The methodology should be strengthen appropriately, some parts are difficult to follow. For example:

- In section 2.2.1, how to correct the effect of local incident angle in Sentinel-1 imagery? The annual composites come from the average of the ascending and descending scenes, why not just ignore ascending and descending orbits in the median compositing?

Response #2: We have now added an explanation about the purpose of the local incident angle correction in Sentinel-1. In addition, we have now added the Google Earth Engine code that we used for generating the Sentinel-1 composites. In this way, the reader can explore the local incident angle correction in more detail (line 94):

"*The correction of Sentinel-1 data for the local incident angle uses SRTM Digital Elevation Data Version 4 to reduce terrain-induced variations in radar backscatter. The correction was applied to daily Sentinel-1 scenes. The code for the correction of the local incident angle and the generation of the Sentinel-1 composites can be found in the Code availability section (Descals, 2021).*"

The revised manuscript also explains why we computed the average for the ascending and descending orbits, separately, and then make the average of these two (line 97):

"*The daily Sentinel-1 images were aggregated annually from 2016 to 2021 using the median for ascending and descending orbits separately. Temporal information, such as seasonal variations in spectral backscatter, was not extracted from the Sentinel-1 time series. The final annual composites represent the average of these two orbit composites. Aggregating the orbits separately addresses imbalances in the number of scenes between orbits, which could otherwise introduce potential terrain-induced artifacts if one orbit prevails.*"

We have also included more clarifications about the methodology. For instance, as suggested by the reviewers, we justified why we used the NDWI, detailed the distinctions between smallholder and industrial oil palm, and included more details about the sampling design used for the validation.

- In section 2.2.2, I suggest add some descriptions about the "closed-canopy industrial oil palm" and "closed-canopy smallholder oil palm", what are the significant differences between these two oil palms?

Response #3: Thank you for the suggestion. Initially, we omitted the description because this was explained in a previous study, but we agree with the reviewer that this current paper should also differentiate between the two classes. We have now added the description of "closed-canopy industrial oil palm" and "closed-canopy smallholder oil palm" (line 115):

"*In this study, we adopted the definitions of industrial and smallholder oil palm plantations from Descals et al., 2021. Industrial plantations typically span several thousand hectares, with uniform palm age and well-defined, often rectangular boundaries (Fig. A1). These plantations feature dense networks of roads or canals, designed during initial development of the plantation to optimize harvesting. On flat terrain, the roads are arranged in a rectilinear grid, while on hilly areas, they tend to curve. In contrast, smallholder plantations are usually less than 25 ha, though this threshold varies by country. Compared to industrial plantations, smallholder plantations are less organized and have more diverse palm ages, forming a mosaic landscape mixed with other land uses. Large clusters of smallholder plantations have sparser trail networks than industrial ones.*"

Please note that we added Fig. A1, which illustrates the differences between these two oil palm classes.

[Figure]

*Figure A1: Sentinel-2 true color composite depicting industrial and smallholder plantations in a region in Riau province, Indonesia.*

In section 2.2.5, authors proposed to use the time-series NDWI to identify the planting years, and gave an example. However, I have two concerns as: 1) the reasonableness of your choice of NDWI, it must be explained. Does it also apply in case of conversion of cropland to oil palm? 2) You've labeled vegetation, clouds, and bare land with random forest classification

over each Landsat observations, but how do you account for the effects of classification error? If an area has gone through "deforestation - bare land - grassland/cropland - oil palm", how can the year of planting be determined?

Response #4: We have now included a rationale for using NDWI (line 197):

"*We selected NDWI because it is less noisy than indices relying on visible spectrum bands. NDWI uses SWIR and NIR bands, which can penetrate thin clouds and are less affected by atmospheric conditions like water vapor, which is typically high in the tropics.*"

Our methodology estimates the planting year of current oil palm plantations; this methodology works independently on the land cover before the current plantation. The Landsat observations used for estimating the planting year correspond to observations during the young stage of the current plantation. Landsat observations prior to the establishment of the current plantation do not influence the planting year mapping. We now rephrased a key sentence in the methods section (lines 207):

"*The purpose of the Random Forest classification was to detect the last Landsat observations depicting bare land, which reflect the preparation of land for the current oil palm plantation, as of 2021.*"

Please note that the planting year we report does not necessarily reflect the year when oil palm was firstly planted; the layer reflects the planting year of the current oil palm plantation, which can be a rotation of an existing oil palm plantation.

The manuscript highlights that we estimated the planting year of the *current* oil palm plantation, not the year when oil palm was firstly established (line 224):

"*Our estimation reflects the planting year of the oil palm plantation detected between 2016 and 2021. Our definition of the planting year may not coincide with the year when oil palm was established for the first time, which could be substantially earlier than our estimate. This scenario arises in plantations that undergo a rotation, wherein oil palm is clear-cut and subsequently replanted with new oil palm seedlings.*"

We also highlight this in the abstract:

"*The planting year indicates the year of establishment for the current oil palm plantation, as of 2021, either newly planted or replanted oil palm in an existing oil palm plantation.*"

The random forest model is a component of the algorithm used to calculate the planting year. While this model has classification errors, we did not directly assess its performance using accuracy metrics like Overall Accuracy, Producer's Accuracy, or User's Accuracy. Instead, we evaluated the validity of the entire algorithm by comparing the oil palm planting year derived from the model with field data, which is the primary variable of interest. If the random forest model performed poorly, this would be reflected as a weak agreement between our oil palm planting year map and the ground truth data.

- In section 2.2.4, the rigor of the validation sample is very clearly stated, but the number of validation samples for oil palm is too sparse (only 973 points). Thus, I suggest increasing the number of oil palm sample points.

Response #5: Thanks for the recommendation. We have now included more sample points. Specifically, we have doubled the number of points for the classes 'smallholder oil palm' and 'industrial oil palm' using a stratified random sampling. All accuracy metrics have been updated accordingly. The accuracy metrics and area estimates do not differ substantially in the revised version compared to the submitted version.

I suggest that the authors give a map of the global spatial distribution of oil palm in 2021 in Section 3.1.

Response #6: Thank you for the suggestion. The Results section now starts with a figure depicting the global oil palm distribution map (Fig. 3). Please note that the 10-meter resolution map is too detailed and data-heavy for global display. For this reason, we included a global density map with lower resolution for efficient visualization and communication of the results.

[Figure]

**Figure 3: Global oil palm density map showing the density of oil palm at a 5 km resolution, derived from the 10 m global oil palm layer for the period 2016-2021.**

The new developed oil palm has been validated to achieve better performance, thus, I suggest that the authors can give some regional, visual comparison results.

Response #7: We have now added a new figure that illustrates the key improvement in the new oil palm layer compared to the previous version. The improvement is highlighted in the main text (line 260):

"*The mapped area is larger than the 2019 global oil palm layer because the updated version includes young oil palm, either from emerging plantations that reached the full canopy closure during the period 2016–2021 or from existing plantations that were replanted before 2019 (Fig. A6).*"

[Figure]

***Figure A6: Comparison between the oil palm dataset developed by Descals et al. (2021) for 2019 and the oil palm extent layer developed in this study for 2016-2021. Black pixels indicate oil palm detected in both datasets, orange pixels represent newly detected oil palm in this study, and blue pixels correspond to areas detected in 2019 but not included in this study. The orange pixels primarily correspond to young oil palm omitted in the previous version, while the blue pixels mostly consist of false positives in coconut-growing regions.***

We have also included this comparison layer (displayed in Fig. A6) in the Google Earth Engine app. This allows users to closely inspect regional differences and improvements:

https://ee-globaloilpalm.projects.earthengine.app/view/global-oil-palm-planting-year-1990-2021

If possible, I would also suggest a global map of the age distribution of oil palms is presented in the manuscript, it would be very interesting!

Response #8: A global map at 10m resolution cannot be effectively displayed because the level of detail is too fine to be properly visualized at a global scale. When zoomed out to view the entire globe, the high-resolution details are lost, making it impossible to distinguish meaningful information; individual 10m pixels are too small to be represented accurately on global scale. For global visualizations, coarser resolutions (>1km) are best suited; this is why, in the revised version, we included the density map of oil palm plantations at 5km resolution; it is a resized version of the 10m oil palm extent. Resizing the age map is not that straightforward as the oil palm age information would be lost, especially in areas with heterogeneous planting years. For this reason, we opted for displaying Fig. 4, which follows the global density map in Fig. 3 and depicts a detail of the oil palm age for an oil palm hotspot region in Indonesia.

I agree with the comment in the Community Comment (CC1) as: the global oil palm area discrepancy between your dataset of 23.98 Mha and the FAO estimate about 30 Mha should be explained in the manuscript

Response #9: We have now added a new figure that compares our oil palm area estimates with the statistics reported by FAO and USDA at the country level. The Results section now presents the comparison with FAO and USDA statistics (line 271):

"*Our area estimates also align with national statistics for oil palm harvested areas reported by FAO and USDA (Figure A7). The largest discrepancy occurred in Nigeria, where we estimated 0.38 ± 0.13 Mha, compared to the 4.86 Mha and 3.00 Mha reported by FAO and FAS-USDA, respectively. This difference may result from the inclusion of semi-wild oil palms in the FAO and USDA statistics. Semi-wild oil palm, common in West Africa, is mostly omitted in our oil palm layer as these palms typically grow scattered across the landscape, making them difficult to map accurately with Sentinel-1.*"

The revised Discussion section also mentions the differences between our oil palm area estimates and the statistics from FAO and USDA (lines 373):

"*Subsistence-level palm oil in Africa could add millions of hectares; areas of these unaccounted traditional oil palm plantations were estimated to be 6.66 Mha in Africa in 2013 (Carrere, 2010). The presence of unaccounted semi-wild oil palms likely explains the ~4.5 Mha discrepancy between our area estimates and FAO's oil palm area in Nigeria, as well as the difference between our global oil palm mapped area (23.98 Mha) and the FAO's reported global harvested area (29.62 Mha) for 2021. Despite this discrepancy, the comparison with official statistics supports the validity of our oil palm extent layer, as our area estimates closely align with the FAO and USDA-reported oil palm areas in other countries.*"

[Figure]

*Figure A7: Oil palm area for the 10 highest producing countries according to the dataset presented in this study. The bars depict the oil palm area for 2021 according to official statistics (FAO and USDA), the blue circles represent the mapped oil palm area using the deep learning model, and the red line shows our oil palm area estimate with a 95% confidence interval.*

---

## Author Comment (AC3)

**Response to Referee #2**

Oil palm has received substantial mapping efforts, but up-to-date and accurate maps detailing both the extent and age of oil palm plantations are essential for monitoring impacts and informing concurrent debates. This study achieved global 10-meter resolution mapping of oil palm and confirmed plantation years from 1990 to 2021. Overall, the manuscript is well-written, but there are still areas that could be improved:

Response #1: Thank you for the positive assessment of the quality of the manuscript. We have now addressed your comments. Please see below a detailed response to all the comments.

1.  In line 22, "We found oil palm plantations covering a total mapped area of 23.98 Mha," while in line 38, "30 Mha in 2022" is mentioned. It is recommended to explain the differences between these two data points in the subsequent discussions.

Response #2: Thank you for noticing this. We have now added a new figure that compares our oil palm area estimates with the statistics reported by FAO and USDA at the country level. We present these results and explain the differences with FAO and USDA statistics in the Results section (lines 271):

"*Our area estimates also align with national statistics for oil palm harvested areas reported by FAO and USDA (Figure A7). The largest discrepancy occurred in Nigeria, where we estimated 0.38 ± 0.13 Mha, compared to the 4.86 Mha and 3.00 Mha reported by FAO and FAS-USDA, respectively. This difference may result from the inclusion of semi-wild oil palms in the FAO and USDA statistics. Semi-wild oil palm, common in West Africa, is mostly omitted in our oil palm layer as these palms typically grow scattered across the landscape, making them difficult to map accurately with Sentinel-1.*"

[Figure]

*Figure A7: Oil palm area for the 10 highest producing countries according to the dataset presented in this study. The bars depict the oil palm area for 2021 according to official statistics (FAO and USDA), the blue circles represent the mapped oil palm area using the deep learning model, and the red line shows our oil palm area estimate with a 95% confidence interval.*

The revised Discussion section also mentions the differences between our oil palm area estimates and the statistics from FAO and USDA (line 373):

"*Subsistence-level palm oil in Africa could add millions of hectares; areas of these unaccounted traditional oil palm plantations were estimated to be 6.66 Mha in Africa in 2013 (Carrere, 2010). The presence of unaccounted semi-wild oil palms likely explains the ~4.5 Mha discrepancy between our area estimates and FAO's oil palm area in Nigeria, as well as the difference between our global oil palm mapped area (23.98 Mha) and the FAO's reported global harvested area (29.62 Mha) for 2021. Despite this discrepancy, the comparison with official statistics supports the validity of our oil palm extent layer, as our area estimates closely align with the FAO and USDA-reported oil palm areas in other countries.*"

2. It is recommended to add descriptions for "closed-canopy industrial oil palm" and "closed-canopy smallholder oil palm". Additionally, using comparative charts to illustrate the differences between these land cover categories both qualitatively and quantitatively is advised, enabling readers to better understand the distinctions between these categories."

Response #3: Thank you for the suggestion. Initially, we omitted the description because this was explained in a previous study, but we agree with the reviewer that this current paper should also differentiate between the two classes. We have now added the description of "closed-canopy industrial oil palm" and "closed-canopy smallholder oil palm" in line 115:

"*In this study, we adopted the definitions of industrial and smallholder oil palm plantations from Descals et al., 2021. Industrial plantations typically span several thousand hectares, with uniform palm age and well-defined, often rectangular boundaries (Fig. A1). These plantations feature dense networks of roads or canals, designed during initial development of the plantation to optimize harvesting. On flat terrain, the roads are arranged in a rectilinear grid, while on hilly areas, they tend to curve. In contrast, smallholder plantations are usually less than 25 ha, though this threshold varies by country. Compared to industrial plantations, smallholder plantations are less organized and have more diverse palm ages, forming a mosaic landscape mixed with other land uses. Large clusters of smallholder plantations have sparser trail networks than industrial ones.*"

We have also added Fig. A1, which illustrates the differences between these two oil palm classes.

[Figure]

*Figure A1: Sentinel-2 true color composite depicting industrial and smallholder plantations in a region in Riau province, Indonesia.*

3. In line 92, it is suggested to briefly explain the data organization format, whether temporal information was used, whether monthly synthesis was employed, and if data stacking akin to multi-channel image data was implemented when inputting into DeepLabv3+.

Response #4: The revised Methods section now clarifies that only the annual aggregates were used, and no temporal information was extracted from the Sentinel-1 time series (line 97 in Section 2.2.1 Sentinel-1 compositing).

"*The daily Sentinel-1 images were aggregated annually from 2016 to 2021 using the median for ascending and descending orbits separately. Temporal information, such as seasonal variations in spectral backscatter, was not extracted from the Sentinel-1 time series.*"

We also clarified how the Sentinel-1 data was inputted into the deep learning model (lines 127 in Section 2.2.2 Deep learning classification):

"*Since the original model required an input image with three channels, we stacked the VV and VH spectral images along with a third image filled with zeros. In this way, we could use the existing deep learning model architecture without modification.*"

4. Due to regional distribution differences, it is recommended to increase the number of validation samples to better demonstrate the effectiveness of the method.

Response #5: Thanks for the recommendation. We have now included more sample points. Specifically, we have doubled the number of points for the classes 'smallholder oil palm' and

'industrial oil palm', respectively, using a stratified random sampling. All accuracy metrics have been updated accordingly. The accuracy metrics and area estimates in the revised version do not differ significantly from the submitted version.

5. In line 119, "manually digitized false positives, and subsequently reclassified these commission errors as class 'other'", was it first masking and then checking all categories, followed by manually correcting misclassified information?

Response #6: Thank you for noticing this. It was first masking and then manual correction. This has now been clarified in line 133:

"*[...], we applied two amendments to reduce the occurrence of false positives. First, we masked oil palm pixels that overlapped with the classes 'cropland', 'built-up', 'water bodies', 'herbaceous wetland', and 'mangrove' in the 10-m ESA WorldCover map v200 (Zanaga et al., 2022), given that oil palm is unlikely to be present in these land cover types. Second, we inspected the annual oil palm classification using high-resolution satellite imagery from Google Maps to remove any remaining false positives. We visually identified these false positives and reclassified them as the class 'other'.*"

6. In the figure on line 179, the author uses NDWI to determine the planting year and provides an example. It is recommended to explain the rationale for using NDWI.

Response #7: Thanks for the recommendation. We have now included a rationale for using NDWI (line 197):

"*We selected NDWI because it is less noisy than indices relying on visible spectrum bands. NDWI uses SWIR and NIR bands, which can penetrate thin clouds and are less affected by atmospheric conditions like water vapor, which is typically high in the tropics.*"

7. In line 228, "only represent the regions where oil palm was found worldwide in this study", did the study detect globally and then filter out these current regions, or was it based on previous research?

Response #8: The oil palm classification was mostly applied to regions delineated in a previous study, except few regions identified in this current study. We have now added this information in the main text (line 148):

"*These 100 x 100 km grid cells represent the regions where oil palm was identified in the 2019 version, presented in Descals et al., 2021, as well as grid cells where oil palm was omitted in the previous version (Fig. A4). The new regions mainly include an oil palm hotspot in the state of Andhra Pradesh in India, industrial plantations in the Congo basin, and scattered plantations in Thailand and Central and South America.*"

8. In Section 3.1, it is suggested to first present the global oil palm distribution map obtained in this study.

Response #9: Thank you for the suggestion. The Results section now starts with a figure depicting the global oil palm distribution map (Fig. 3). Please note that the 10-meter resolution map is too detailed and data-heavy for global display. For this reason, we included a global density map with lower resolution for efficient visualization and communication of the results.

[Figure]

*Figure 3: Global oil palm density map showing the density of oil palm at a 5 km resolution, derived from the 10 m global oil palm layer for the period 2016-2021.*

9. In Section 3.3, consider increasing the number of sample points to assess validity further.

Response #10: We have now included more sample points. Specifically, we have doubled the number of points for the classes 'smallholder oil palm' and 'industrial oil palm', respectively, using a stratified random sampling. All accuracy metrics have been updated accordingly. The accuracy metrics and area estimates in the revised version do not differ significantly from the submitted version.

It is recommended to add comparative results with previous studies in Section 3.3 to visually demonstrate the effectiveness of the current research.

Response #11: We now compared our oil palm extent with official statistics from the FAO and USDA and with a previous study in Indonesia (Gaveau et al., 2022). Additionally, we assessed the improvements in this study over the previous version (Descals et al., 2021). For oil palm age, we validated our dataset using field measurements and inspection of Landsat time series. We did not compare our results with oil palm age layers from other studies for two reasons: (1) Other studies define the planting year differently. For example, Gaveau et al. (2022) defines the planting year as the year of conversion to oil palm, whereas we include oil palm rotations as the planting year. These differences introduce inconsistencies in the comparison, potentially undermining the accuracy of our age layer. (2) No reliable global reference dataset exists for oil palm age; other studies contain errors, making comparisons likely to produce misleading results. This is the reason why we produced a validation dataset extracted from Landsat to directly evaluate the accuracy of the planting year layer.